# Brain-specific lipoprotein receptors interact with astrocyte derived apolipoprotein and mediate neuron-glia lipid shuttling

Jun Yin[1], Emma Spillman[1,5], Ethan S. Cheng[1], Jacob Short [1], Yang Chen[1], Jingce Lei[1], Mary Gibbs[1], Justin S. Rosenthal[1], Chengyu Sheng [1,6], Yuki X. Chen[2,3], Kelly Veerasammy[2,3], Tenzin Choetso[2,3], Rinat Abzalimov[2], Bei Wang [4], Chun Han [4], Ye He[2] & Quan Yuan [1✉]

Lipid shuttling between neurons and glia contributes to the development, function, and stress responses of the nervous system. To understand how a neuron acquires its lipid supply from specific lipoproteins and their receptors, we perform combined genetic, transcriptome, and biochemical analyses in the developing *Drosophila* larval brain. Here we report, the astrocyte-derived secreted lipocalin Glial *Lazarillo* (GLaz), a homolog of human Apolipoprotein D (APOD), and its neuronal receptor, the brain-specific short isoforms of *Drosophila* lipophorin receptor 1 (LpR1-short), cooperatively mediate neuron-glia lipid shuttling and support dendrite morphogenesis. The isoform specificity of LpR1 defines its distribution, binding partners, and ability to support proper dendrite growth and synaptic connectivity. By demonstrating physical and functional interactions between GLaz/APOD and LpR1, we elucidate molecular pathways mediating lipid trafficking in the fly brain, and provide in vivo evidence indicating isoform-specific expression of lipoprotein receptors as a key mechanism for regulating cell-type specific lipid recruitment.

[1] Dendrite Morphogenesis and Plasticity Unit, National Institute of Neurological Disorders and Stroke, National Institutes of Health, Bethesda, MD, USA. [2] The City University of New York, Graduate Center-Advanced Science Research Center, New York, NY, USA. [3] The City College of New York, CUNY, New York, NY, USA. [4] Weill Institute for Cell and Molecular Biology and Department of Molecular Biology and Genetics, Cornell University, Ithaca, New York, USA. [5] Present address: Department of Neurosciences, University of California, San Diego, San Diego, CA, USA. [6] Present address: Department of Pharmacology, School of Basic Medical Sciences, Nanjing Medical University, Nanjing, China. ✉email: quan.yuan@nih.gov

Lipid trafficking and homeostasis are critical for the development and maintenance of the nervous system. These processes are mediated by a large set of molecular carriers shuttling a diverse group of lipid cargos in and out of designated cell types and cellular compartments[1,2]. In the central nervous system (CNS), lipid homeostasis heavily relies on neuron-glia cross talk[3]. Studies in mammalian systems have indicated that, besides direct anatomical interactions, glia also supply neurons with metabolic substrates, antioxidants, and trophic factors through secretion[4–6]. Intriguingly, apolipoproteins are among the most abundant secretory factors that are produced and released by mammalian astrocytes[6,7], a group of glial cells with complex morphology and highly branched structures that are intimately associated with synapses[6,8], suggesting a critical role for glia-derived lipoprotein and their lipid cargos in synapse formation and function[2,9]. This notion is supported by studies in cultured mammalian CNS neurons, where glia-derived cholesterol and phospholipids are essential for synaptogenesis[9–11]. In addition, recent findings in the Drosophila system also indicate essential functions of glia in synapse formation and neurotransmission[12–15], although the link between neuron-glia lipid transport and synaptic function has yet to be established.

Characterized by their high metabolic rate and elaborate morphology, neurons require a continuous lipid supply throughout their lifetime. How lipoproteins and their receptors mediate neuron-glia lipid shuttling to meet that demand has been a long-standing question in the neurobiology field. Numerous studies over the past decades have demonstrated the critical functions of CNS lipid trafficking in synapse development and cognitive functions[3,16,17]. In the mammalian system, deficiencies in either apolipoproteins or their receptors lead to both structural and functional deficits in the brain. For example, Apolipoprotein E (ApoE) delivers cholesterol, amyloid-β, and other hydrophobic molecules to neurons through its interaction with Very Low-Density Lipoprotein Receptor (VLDLR) and Apolipoprotein E Receptor 2 (ApoER2)[16,18]. While the Apolipoprotein E (ApoE) knockout mice display significantly reduced dendrite size and synapse number as well as impaired learning and memory[19], both VLDLR and ApoER2 knockout animals also show deficits in cerebellar morphology and impaired contextual fear conditioning and long-term potentiation[20,21]. Genetic studies of other lipid transport proteins and receptors, including APOD, Niemann–Pick Type C (NPC), Low-Density Lipoprotein Receptor (LDLR), and Low-density lipoprotein Receptor-related Protein 1 (LRP1), further support the importance of lipid trafficking in the proper development and function of the nervous system[22–25].

Due to the diversity of lipid transport proteins and lipoprotein receptors, as well as the complexity of their tissue- and cell-specific distributions, cellular and molecular mechanisms underlying neuron-glia lipid shuttling have not been well characterized in vivo under physiological conditions[3,26]. Recent findings in Drosophila highlight the protective functions of neuron-glia metabolic coupling in neurons experiencing oxidative stress or enhanced activity, demonstrating how neurons transfer lipids into glia for detoxification and storage[27–30]. Similarly, observations made in the mammalian system also provided evidence illustrating fatty acid (FA) transport into astrocytes mediated by ApoE and the importance of neuronal lipid clearance[31,32]. In contrast, much less is known about how neurons acquire lipid cargos from glia-derived lipoproteins under normal conditions, especially during development, when neurite outgrowth and synapse formation produce a high lipid demand. We sought to fill this gap by determining the functional significance and regulatory mechanisms underlying neuronal lipid uptake using the Drosophila larval brain as a model system.

In Drosophila, the Apolipoprotein B (ApoB) family lipoprotein apolipophorin (apoLpp) is a major hemolymph lipid carrier and has two closely related lipophorin receptors (LpRs), LpR1 and LpR2, both of which are homologs of mammalian LDLR family proteins[33–35]. Notably, Drosophila LpRs have multiple isoforms produced by alternative splicing and differential promoter usage[33]. In the fly imaginal disc and oocyte, long isoforms of LpRs (LpR-long) interact with lipid transfer particles (LTP, Apoltp) and mediate endocytosis-independent neutral lipid uptake, while short isoforms of LpRs (LpR-short) neither bind to LTP, nor mediate lipid uptake in these peripheral tissues[33,34]. In contrast, our previous genetic studies revealed specific expression of LpR-short in larval ventral lateral neurons (LNvs), a group of visual projection neurons, and its functions in supporting dendrite development and synaptic transmission[36]. This observation is validated by a recent study performed in a cultured Drosophila neuronal cell line, where the LpR-dependent lipid uptake was directly visualized using fluorescently labeled ApoLpp[37]. Is there a functional significance behind the isoform-specific expression of LpRs? How do short isoforms of LpRs mediate lipid uptake in neurons? These are the questions we aim to address.

In this study, we focus on the LpR1 gene and uncover its isoform-specific expression in neurons and its functions in regulating brain lipid content. Through systematic genetic and biochemical analyses, we identify Glial Lazarillo (GLaz), an astrocyte-derived secreted lipocalin and a homolog of human APOD, as a binding partner for the brain-specific LpR1-short and reveal their cooperative functions in supporting dendrite morphogenesis, synaptic transmission, and lipid homeostasis in the developing larval brain. In adult Drosophila, GLaz/APOD is found in CNS glia and has been shown to regulate stress resistance and contribute to longevity[38,39]. Recent studies also demonstrated that GLaz mediates neuron to glia lipid transfer and facilitates neuronal lipid clearance[27,28]. By identifying GLaz's function in neural development and its direct interaction with LpR1, we not only uncover a pair of molecular carriers mediating neuron-glia lipid shuttling in the Drosophila CNS but also present evidence supporting isoform-specific expression as a key mechanism for regulating the tissue distribution and ligand specificity of neuronal lipoprotein receptors. This in turn leads to the stage- and cell-type-specific regulation of lipid uptake.

## Results

**Isoform-specific expression and activity-dependent regulation of LpR1 in LNv neurons.** To determine how neuronal lipoprotein receptors mediate lipid uptake, based on findings from previous studies, we focus our genetic and functional analyses on the Drosophila LpR1 gene, which expresses in the larval CNS and is required for LNvs' dendrite development and synaptic functions[36]. Sequence analyses revealed that LpR1 shares conserved functional domains with the mammalian LDLR family proteins, including LDLR receptor type A modules (LA), EGF modules, and ß-propellers (Fig. 1a). Multiple isoforms of LpR1 have been detected in Drosophila cDNAs with distinct tissue distributions and properties[33,34]. They are generally categorized as either long (LpR1-long) or short isoforms (LpR1-short) (Supplementary Fig. 1 and Fig. 1b). Compared to the long isoforms, LpR1-short isoforms have a shorter N-terminal domain and one less LA module (Fig. 1a, b).

Our RNA-seq analyses revealed isoform-specific expression of LpR1 in LNvs (Fig. 1b)[36]. While both long and short isoforms of LpR1 are detected in the ModEncode larval CNS RNA-seq data set (ModEncode ID: 4658)[36,40]. LNv-specific libraries only show detectable reads from LpR1-short specific exons (exon 5–6) and the LpR1-long specific reads (exon 1–4) are absent, indicating that

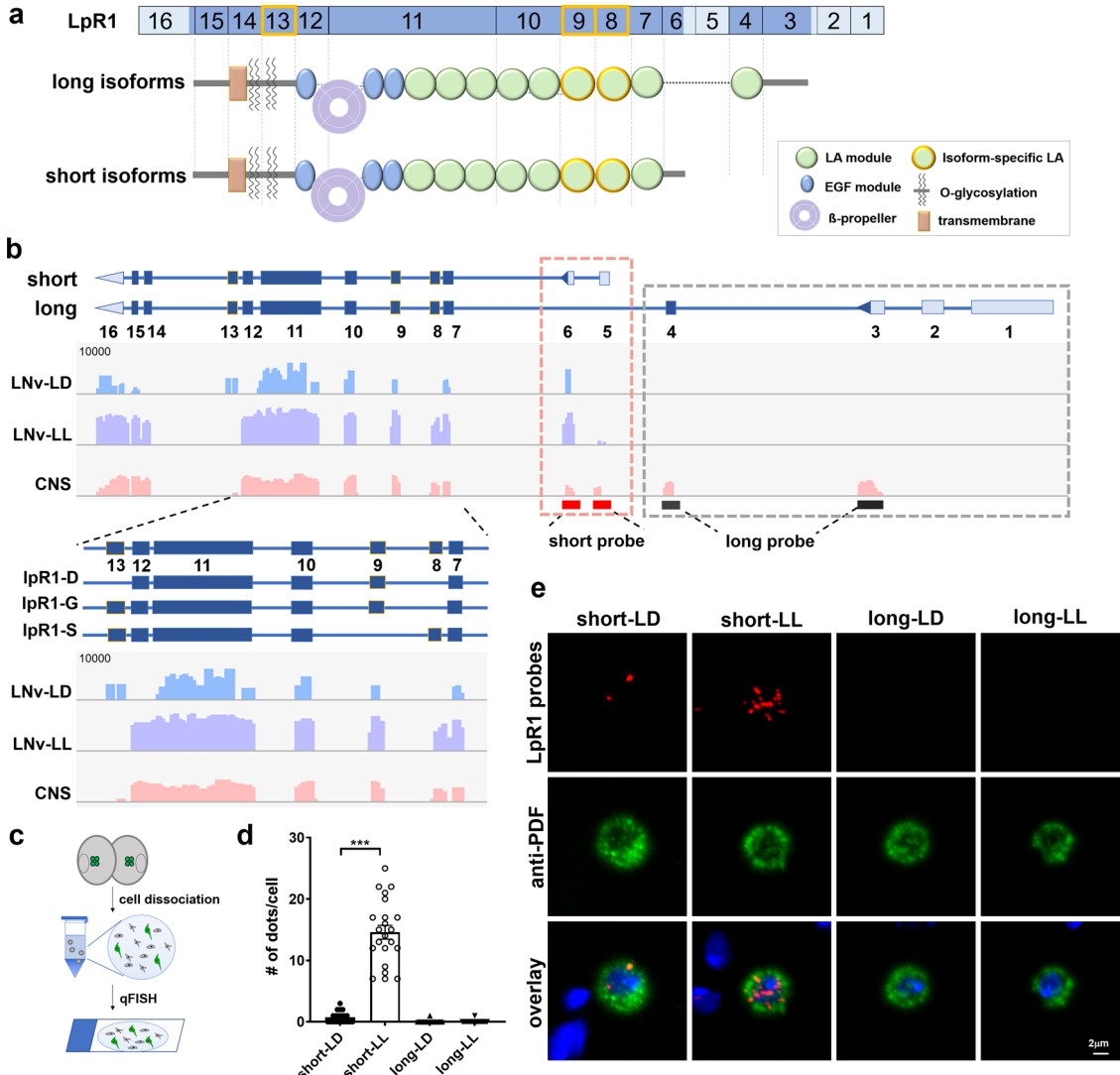

**Fig. 1 Isoform-specific expression of LpR1 is detected in the LNvs. a** A schematic illustration of long and short isoforms of the LpR1 receptor. Top: exon organization of the *LpR1* gene. Untranslated regions are in light blue, coding exons are in dark blue. Flexible exons are outlined with orange borders. Bottom: protein domains of LpR1. Long and short-isoforms differ in their N-terminal regions (gray bars) and the numbers of LA modules (green circles). **b** Exon-mapping of RNA-seq data reveals that only *LpR1-short* isoforms are expressed in LNvs in both light: dark (LD) and constant light (LL) conditions. Top: schematic representation of the *LpR1* gene structure. Bottom: Reads from LNv-specific and CNS RNA-seq libraries mapped to each exon of the *LpR1* gene. The lower panel shows a magnified view of the flexible exons and the corresponding *LpR1-short* isoforms. Untranslated regions are in light blue, coding exons are in dark blue, and flexible exons are outlined with orange borders. Reads number is log scaled. Top Right: isoform-specific FISH probes targeting *LpR1-short* (red bars) and *LpR1-long* (black bars) specific sequences. **c** A diagram illustrating the workflow for qFISH with dissociated brain cells. **d** Only *LpR1-short* is expressed in LNvs and is upregulated by excessive visual input in the LL condition. Quantifications of the transcript levels (# dots/cell by qFISH) of *LpR1-short* and *LpR1-long* in LNvs from LD and LL conditions are shown. Data are presented as mean values +/− SEM. Statistical significance was assessed by one-way ANOVA with Tukey's post hoc test. ANOVA: $p < 0.0001$, $F = 143.3$, df = 78; short-LD/short-LL: $p < 0.0001$; long-LD/long-LL: $p > 0.9999$. $n = 20, 22, 20$, and 20 for short-LD, short-LL, long-LD, and long-LL, respectively. $n$ represents individual LNvs. ***$p < 0.001$. **e** Representative confocal images of qFISH results on single LNvs are shown (observed in at least 20 individual cells). *LpR1-short* transcripts (red dots) are detected in LNvs (stained by anti-PDF in green). Nuclei are stained by DAPI (blue).

LNvs specifically express *LpR1-short* isoforms (Fig. 1b)[33,34]. In addition, because the reads from flexible exons 8, 9, and 13, corresponding to three different *LpR1-short* isoforms, are all detected in LNv-specific RNA-seq libraries, we conclude that LNvs potentially express all three short isoforms of *LpR1* (Fig. 1b).

To validate the short isoform-specific expression of *LpR1* in LNvs, we performed quantitative fluorescent in situ hybridization (qFISH) on acutely dissociated LNvs using *LpR1-short* and *LpR1-long* specific probes (Fig. 1c). qFISH analyses confirmed the specific expression of *LpR1-short* in LNvs, and also showed a significant increase in *LpR1-short* transcript levels induced by the constant light condition (LL), in which LNvs receive chronically elevated synaptic input[41] (Fig. 1d, e). In contrast, *LpR1-long* transcripts have no LNv expression in the regular light: dark (LD) condition and no change in the LL condition, consistent with the RNA-seq results. These results strongly support that only the *LpR1-short* isoforms are expressed in LNvs and synaptic activity selectively induces the up-regulation of the *LpR1-short* transcripts.

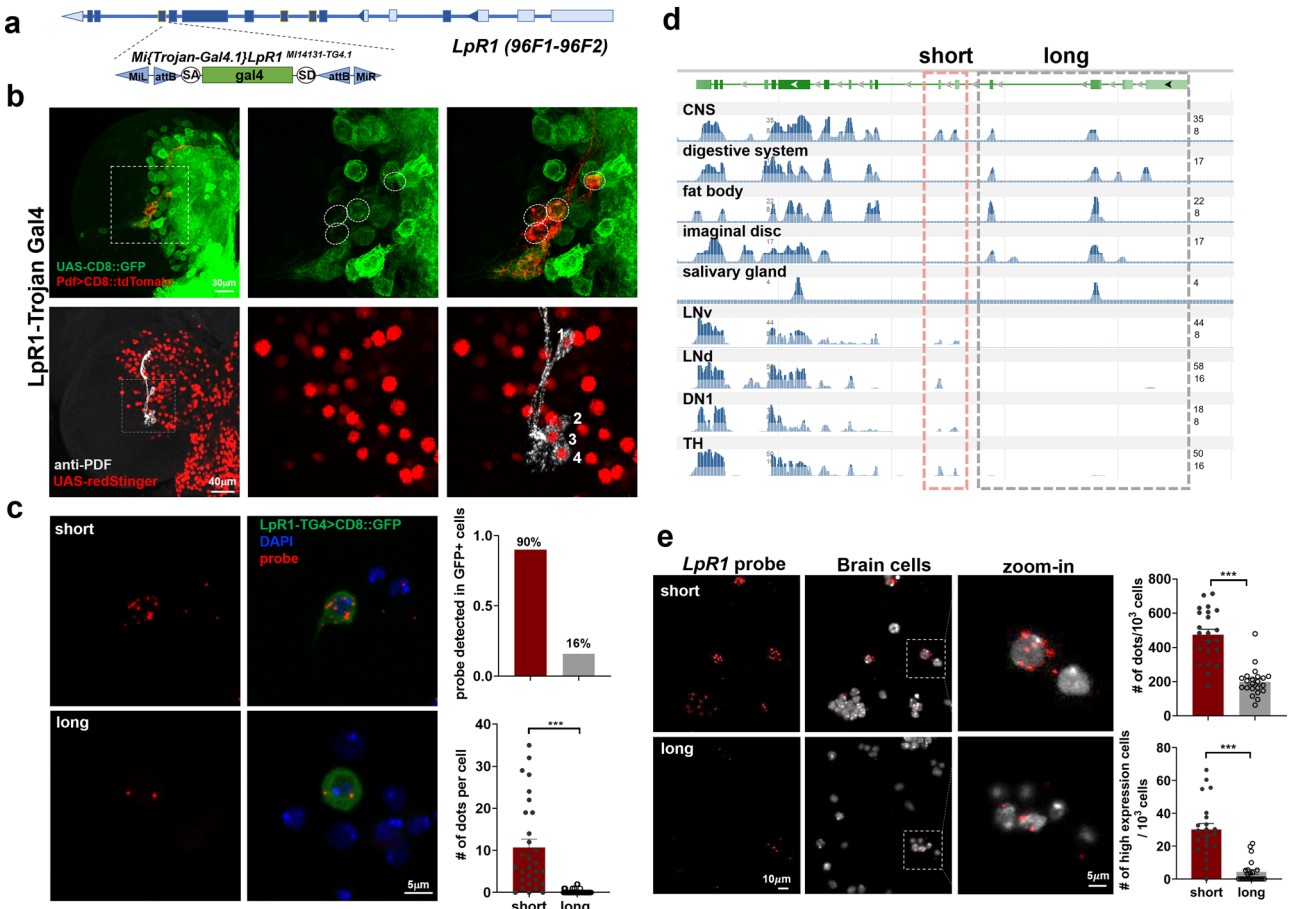

**Fig. 2 LpR1-short exhibits brain-specific expression and is the dominant isoform in the larval brain. a**, **b** LpR1 is widely distributed in the larval brain. **a** Schematic diagram of a Trojan-Gal4 insertion at the 3′ end of the *LpR1* gene (LpR1-TG4), which represents the expression of all *LpR1* isoforms. **b** The broad distribution of LpR1 in the larval brain is detected by the LpR1-TG4 driving CD8::GFP (top) or redStinger (bottom) (observed in at least 10 brains). The expression of LpR1 in LNvs is confirmed by co-labeling using either Pdf-LexA driving tdTomato (red, circles in top panels) or anti-PDF antibody (gray, numbers in bottom panels). **c** *LpR1-short* is the dominant isoform expressed in the larval brain. Representative confocal images (left, probe in red and GFP in green) and quantifications (Right) from qFISH analyses on *LpR1*-expressing brain cells are shown. In the bottom graph, data are presented as mean values +/− SEM. $p < 0.0001$, $t = 6.143$, df = 78, by two-tailed Student's $t$-test. $n = 30$ and 50 for short and long. $n$ represents individual GFP positive cells. ***$p <$ 0.001. **d** *LpR1-short* expression is detected in CNS and four types of adult neurons, but not in the peripheral tissues, where only *LpR1-long*-expression is detected. Tissue-specific and neuron-specific RNA-seq displays reads from isoform-specific exons (short-red frame, long-gray frame, reads number log base 2 scaled). **e** *LpR1-short* is more abundant than *LpR1-long* in the larval brain. Representative confocal images (Left, probe: red; DAPI: gray) and quantifications (Right) from qFISH analyses on dissociated brain cells are shown. Compared to *LpR1-long*, *LpR1-short* shows higher total transcript levels in brain cells (# of dots/1000 cells) and the total number of cells with high expression (# of high expression cells/1000 cells). ≥4 dots/cell were counted as high expression. Data are presented as mean values +/− SEM. Statistical significance was assessed by a two-tailed Student's $t$-test. Top: $p < 0.0001$, $t = 7.368$, df = 42. Bottom: $p < 0.0001$, $t = 6.593$, df = 42. $n = 22$ for both groups. $n$ represents the individual quantified image. ***$p < 0.001$.

**Short isoforms of LpR1 are brain-specific and the predominant isoform expressed in neurons**. To examine the general expression pattern of LpR1, including all long and short isoforms, in the larval brain, we obtained the LpR1-MI14131-TG4.1 line (LpR1-TG4), which contains a Gal4 element inserted in the intronic region between exons 12 and 13 and is under the control of the endogenous LpR1 promoter[40,42] (Fig. 2a). When crossed with a membrane-targeted CD8::GFP or a nuclei marker, redStinger, the LpR1-TG4 revealed the wide distribution of LpR1 in the larval brain, including its expression in LNvs (Fig. 2b), supporting the initial RNA-seq results. In addition, the expression of redStinger driven by LpR1-TG4 does not overlap with Repo, a glial marker (Supplementary Fig. 2a). This is consistent with the data from a recent single-cell RNA-seq study in the larval CNS, which indicates that LpR1 transcripts are mostly found in differentiated larval neurons[43] (Supplementary Fig. 2b).

We examined the expression level for *LpR1-short* and *LpR1-long* in LpR1-expressing cells, by performing qFISH analyses on dissociated brain cells labeled by the LpR1-TG4 driving GFP (Fig. 2c). We detected *LpR1-short*-expression in over 90% of the GFP positive cells, but only 16% of the GFP positive cells show *LpR1-long*-expression (Fig. 2c, top). In addition, in individual cells with detectable GFP expression, *LpR1-short* probes show a higher average count than the cells labeled with *LpR1-long* probes (Fig. 2c, bottom).

To analyze the isoform-specific expression of *LpR1* in other neuron types and larval tissues, we then performed exon-mapping on tissue-specific 3rd instar larvae RNA-seq data from NCBI GEO database (GEO accession numbers are included in Methods). Interestingly, while the CNS library shows both *LpR1-short* and *LpR1-long*-expression, in libraries generated using peripheral tissues, including the digestive system, fat body, imaginal disc, and salivary glands, only reads specific for *LpR1-*

*long* were detected (Fig. 2d). In addition, we analyzed previously published neuron-specific RNA-seq libraries that contain expression profiles of several types of adult neurons, including the LNv, dorsal lateral neuron (LNd), dorsal neuron 1 (DN1), and dopaminergic neurons (TH)[44]. The results indicate that all four types of adult neurons express *LpR1-short* with little to undetectable levels of *LpR1-long* transcripts (Fig. 2d). Together, our analyses on the transcriptome data indicate that the short isoforms of *LpR1* are brain/CNS-specific.

To directly examine and quantify the overall expression level for *LpR1-short* and *LpR1-long* in the brain and confirm the results from the RNA-seq analyses, we performed qFISH in dissociated 3rd instar larval brain cells. We found that *LpR1-short* has significantly higher expression than *LpR1-long*, reflected by quantifications of both the averaged probe signal and the highly expressed cell number (Fig. 2e). In combination with the transcriptome analyses and previous findings[34,45], our qFISH results indicate that *LpR1-short* is the dominant isoform of *LpR1* expressed in the brain, supporting the isoform-specific distribution of LpR1 receptor in either peripheral tissues or CNS.

**LpR1-short is required for the proper development of LNv dendrites.** To understand the functional implication of the specific expression of LpR1-short in the brain, we designed experiments to test a pair of long and short isoforms that have identical flexible exons and conserved domains, and only differ in their N-terminus and the number of LA modules. Among all LpR1 isoforms, we found two matching sets: LpR1D (short) and LpR1L (long), LpR1G (short), and LpR1H (long) (Supplementary Fig. 1). Because LpR1D failed to rescue the LpR1 mutant phenotype in our previous study[36], we selected LpR1G and LpR1H as the representative pair we would use to identify the functional distinctions between short and long isoforms.

We generated transgenic lines expressing GFP-tagged LpR1G (LpR1-short-GFP) and LpR1H (LpR1-long-GFP) and first examined their intracellular localization in LNvs by co-expressing either a membrane-targeted red fluorescent marker (CD2::mCherry) or a tdTomato-tagged early endosome marker (Rab5-tdTomato)[46]. Both isoforms localize predominately in the soma and dendritic regions of LNvs, and much less in the axon (Fig. 3a, Supplementary Fig. 3). In addition, we observed that LpR1-short-GFP localizes in vesicular structures that largely overlap with Rab5-tdtomato signals, while LpR1-long-GFP also appears in vesicles but mostly colocalizes with membrane marker CD2::mCherry (Fig. 3a, Supplementary Fig. 3). Similar distinctions were also observed through the expression of HA-tagged LpR1H and LpR1G transgenes that are directly driven by an LNv-specific enhancer sequence (Pdf > LpR1H-HA and Pdf>LpR1G-HA). The transgenes were inserted in the same genomic location by site-specific integration (Supplementary Fig. 3b) and driven by the same enhancer, indicating that the difference in localization we observed between the tagged short and long isoforms is not due to their different levels of expression.

It is possible that our results generated through these overexpression studies may not faithfully represent the endogenous LpR1 distribution. However, the endosomal localization of the LpR1-short-GFP is consistent with previous findings in *Drosophila* and other insects, suggesting that brain-specific short-isoforms of LpRs are endocytic receptors, similar to the mammalian LDLRs[33,47,48].

To parse out functional differences between the short vs. long isoforms of LpR1, we examined their abilities to support LNv dendrite morphogenesis. Our previous results indicate that *LpR1* is required for LNv dendrite development and synaptic function. There is a significant reduction in LNv dendrite volume in the

*LpR1* loss-of-function mutant, which contains a deletion eliminating the entire coding sequence[36]. Here, we performed tissue-specific rescue experiments through the expression of LpR1-short-GFP or LpR1-long-GFP driven by an LNv-specific Gal4 enhancer in the LpR1 mutant background. Through 3D-reconstruction of LNv dendrites and quantification of the total dendrite volume, we found that both LpR1-short and LpR1-long fully rescue the dendrite reduction phenotype in *LpR1* mutants (Fig. 3b). Unexpectedly, the exogenous expression of LpR1-long produced a distinct phenotype, which is characterized by numerous exuberant dendrite branches extending outside of the normal range of the LNv dendritic field and was quantified using 3D-sholl analysis[49] (Fig. 3c).

To understand whether these morphological differences lead to changes in functional connectivity, we then examined the physiological properties of LNvs using calcium imaging experiments. LNvs receive synaptic input from larval photoreceptors and exhibit robust calcium responses upon light stimulation[41]. Using a genetically encoded calcium indicator GCaMP7f[50], we measured light-evoked calcium increases in the axonal terminal region of the LNvs (Fig. 3d). LpR1 mutants showed a significant reduction in the calcium response, which was fully rescued by LpR1-short-expression but remained unchanged in LNvs expressing the LpR1-long isoform (Fig. 3d).

This set of comparisons made between LpR1-long vs -short isoforms reveals strong influences of isoform specificity on the receptor's function. Importantly, replacing the native LpR1-short with the exogenously expressed long isoform in LNvs leads to exuberant dendrite growth and severely dampened physiological responses. Our results suggest that, although both short and long isoforms of LpR1 are capable of recruiting lipids to support dendrite growth, the expression of LpR1-short in neurons is essential for the proper establishment of synaptic connectivity during development.

**CRISPR/Cas9 mediated tissue-specific mutagenesis reveals the specific requirement of LpR1-short in neurons.** Our genetic rescue experiments suggest that isoform-specific expression of LpR1-short in the LNv is required for its proper development and synaptic function. To validate this conclusion using loss-of-function analyses, we performed CRISPR/Cas9-mediated tissue-specific mutagenesis to eliminate either the short or long isoforms of LpR1 in LNvs.

Two sets of gRNA sequences targeting either the long isoform-specific exon 3 or the short isoform-specific exon 6, were selected to make ubiquitously expressed transgenic gRNAs in an expression vector optimized for both germline and somatic mutagenesis[51] (Fig. 4a). The gRNA constructs were inserted into the 2nd chromosome by site-specific integration to generate transgenic lines. Efficiencies of these gRNA constructs were validated using the Cas9-LEThAL assay[52]. Crossing transgenic gRNA males with females of *Act5C-Cas9 lig4*[53] yielded high lethality of male progeny, 77.2% for the short-isoform, and 98% for the long-isoform specific gRNAs, indicative of these gRNAs' high efficacy in producing genome editing events.

Next, we introduced the short-specific and long-isoform-specific gRNA transgenes into flies expressing Cas9 driven by the LNv specific promotor (Pdf-Gal4>UAS-Cas9) and examined the LNv dendrite labeled by CD8::GFP. Our quantifications indicate that CRISPR/Cas9-mediated mutagenesis through LpR1-short specific gRNAs (Short-gRNA) produced a significant reduction of the LNv dendrite volume, while the long isoform-specific gRNAs (Long-gRNA) showed no difference from the control groups, consistent with the results we obtained from the rescue experiments (Fig. 4b, c).

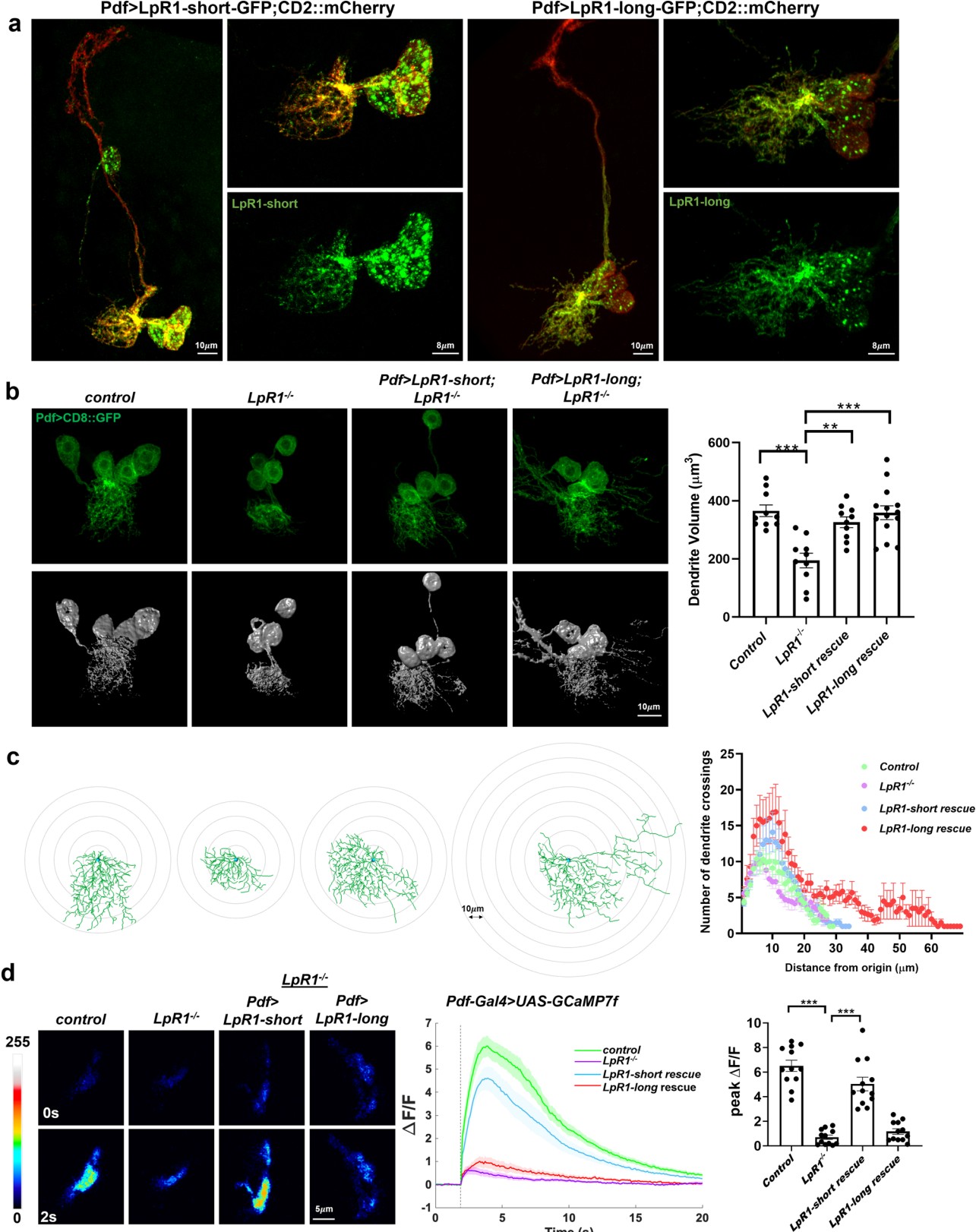

Combined with the expression analyses using RNA-seq and qFISH, our isoform-specific genetic manipulations clearly demonstrate the neuronal-specific expression of LpR1-short and its function in supporting dendrite development and synaptic activity in neurons. These results are complementary to previous findings in *Drosophila* peripheral tissues, where long-isoforms of LpR1 recruit lipids through an endocytosis-independent mechanism and LTP-facilitated cell surface lipolysis[33,47] and demonstrate the functional significance of isoform-specific tissue distributions of the LpR1 receptor.

**Fig. 3 LpR1-short is required for proper development of the LNv dendrite. a** The expression and localization of GFP-tagged LpR1 isoforms in LNvs. LpR1-short-GFP appears to largely localize in vesicles within both soma and dendritic regions, while LpR1-long-GFP is mainly distributed on the cell surface and along the dendrite branches. Representative confocal images of LNvs expressing membrane-targeted CD2::mCherry (red) and GFP-tagged LpR1-short or LpR1-long (green) (observed in at least 10 brains). The whole neuron (left) and the zoomed-in images of soma and dendritic regions (right) are shown. **b** *LpR1* null mutants show a dendrite reduction phenotype in LNvs which is fully rescued by LNv-specific expression of either LpR1-short-GFP or LpR1-long-GFP transgenes. Left: representative confocal images (green, top panels) and 3D reconstructions (gray, bottom panels) of LNv soma and dendritic regions are shown. Right: Quantifications of LNv dendrite volume. Data are presented as mean values +/− SEM. Statistical significance was assessed by one-way ANOVA with Tukey's post hoc test. ANOVA: $p < 0.0001$, $F = 11.67$, $df = 40$; *control/LpR1$^{-/-}$*: $p < 0.0001$; *LpR1$^{-/-}$/LpR1-short rescue*: $p = 0.0019$; *LpR1$^{-/-}$/LpR1-long rescue*: $p < 0.0001$. $n = 10$, 10, 10, and 14 for *control*, *LpR1$^{-/-}$*, *LpR1-short rescue*, and *LpR1-long rescue*. $n$ represents individual larval brain samples. $**p < 0.01$, $***p < 0.001$. **c** Misexpression of LpR1-long leads to abnormal dendrite development in LNvs. 3D-Sholl analysis of the LNv dendrite arbors reveals that the expression of LpR1-long, but not LpR1-short, alters LNv dendrite morphology, represented by exuberant dendritic branches extended beyond the synaptic region. Left: representative projected 3D tracings of the LNv dendrites. Each successive circle represents a 10 μm increase in diameter. Right: Sholl analyses graph. Data are presented as mean values +/− SEM. $n = 10$ for all groups. **d** The *LpR1* mutant displays a severely dampened physiological response in LNvs, which is fully rescued by LpR1-short, but not LpR1-long. Left: Representative frames of the Pdf > GCaMP7f recordings at 0 and 2 s after a light stimulation measured in the axonal terminal region of LNvs. Middle: The average GCaMP signal traces. The shaded area represents SEM. The dashed line represents the light stimulation delivered by a 100-ms light pulse. Right: The quantifications of the peak value of the changes in GCaMP signal induced by light stimulations (ΔF/F). Data are presented as mean values +/− SEM. Statistical significance was assessed by one-way ANOVA with Tukey's post hoc test. ANOVA: $p < 0.0001$, $F = 54.61$, $df = 44$; *control/LpR1$^{-/-}$*: $p < 0.0001$; *LpR1$^{-/-}$/LpR1-short rescue*: $p < 0.0001$; *LpR1$^{-/-}$/LpR1-long rescue*: $p = 0.8290$. $n = 12$ in all groups. $n$ represents individual larval brain samples. $***p < 0.001$.

**LpR1 regulates lipid homeostasis in the larval brain**. To understand the general role of LpR1 in regulating lipid trafficking and homeostasis in the CNS, we examined the lipid content of the fly brain using two different methods: quantification of the lipid droplets and MALDI-TOF MS (Matrix-Assisted Laser Desorption/Ionization time-of-flight mass spectrometry) imaging[54].

Previous immunohistochemistry and electron-microscopy (EM) studies of the *Drosophila* brain have shown that lipid droplets are only found in glia and are strongly influenced by neuron-glia lipid trafficking and metabolic coupling[27–29,55]. Thus, we used the lipid droplet density as a parameter to assess the lipid content in the brain. Lipid droplets in *Drosophila* tissues can be examined by Nile red staining[56], or the expression of a lipid storage droplet-2 fused YFP (Lsd-2-YFP) protein trap[57]. Nile red stains the neutral lipids within the core of the lipid droplets, while Lsd-2 labels their membranes. In the larval CNS, we observed widely distributed lipid droplets labeled by both Lsd2-YFP and Nile red (Fig. 5a). The lipid droplets are also present in the larval optic neuropil region (LON), where LNv dendrites make synaptic contact with the photoreceptor axon (Fig. 5b)[41].

To examine the effect of LpR1 deficiency on the brain lipid content, we performed Nile red staining and specifically quantified the lipid droplet density in the LON region through 3D reconstruction (Fig. 5c). Our analyses revealed a significant reduction in the density of lipid droplets in *LpR1$^{-/-}$* mutants, as compared to the controls (Fig. 5d). Since LpR1 also expresses outside of the CNS, to evaluate the contribution of neuronal expression of LpR1 towards this phenotype, we knocked down LpR1 expression only in neurons using a transgenic RNAi line driven by a pan-neuronal driver Elav-Gal4. Here, we also observed a significant reduction of brain lipid droplet density (Supplementary Fig. 4), suggesting that neuronal expression of *LpR1* has a non-autonomous effect on the glial lipid storage and contributes to the general maintenance of lipid content and homeostasis in the brain.

The main components of the lipid droplet are TAG and cholesteryl esters[58]. To test whether *LpR1* is involved in recruiting and regulating the brain content of other types of lipids, we examined general lipid content in the *LpR1$^{-/-}$* mutant using the MALDI-TOF MS[54]. This approach allows us to visualize many lipid species at once in the adult head sections, including two neutral lipids, TAG[M + Na]$^+$ (C46:3) and DAG[M − OH]$^+$ (C30:1), cholesterol, and five types of major phospholipids, PC[M + H]$^+$ (C32:1, C34:1, C36:2, C36:3), PC-N(CH3)3

[M + H]$^+$ (C30:4, C30:3), PE[M + H]$^+$ (C38:3, C38:4, C38:5), PG[M + Na]$^+$ (C34:2) and PS[M + Na]$^+$ (C32:2, C34:4, C36:5). After quantification of the signal from the brain regions, we found that, compared to the wild-type control, all eight types of lipids showed significant reductions in *LpR1$^{-/-}$* mutants (Fig. 5e, quantifications shown in Supplementary Fig. 5). Together, both Nile red staining and MALDI imaging support the critical function of LpR1 in regulating brain lipid trafficking and homeostasis.

Although MALDI-imaging allowed us to directly visualize a broad spectrum of lipid species, using this approach to analyze the small fly head sections has certain limitations, such as the low spatial resolution, variable detection sensitivity, and potential contamination from non-brain tissues. In addition, both of our analyses revealed a general reduction of lipid contents in the LpR1 mutant, therefore, the specific lipid cargo(s) being transferred by LpR1 within the fly CNS remains unidentified.

**Astrocyte-derived factors regulate LNv dendrite growth**. While the long isoforms of LpR receptors mediate lipid uptake in peripheral tissues through their interactions with LTP and ApoLpp, the potential ligand of the CNS-specific LpR1-short remains unknown[34]. To identify the molecular partner of short isoforms of LpR1, we evaluated various lipid transfer proteins produced within the CNS. Previous studies indicate that, in both vertebrate and invertebrate CNS, neuronal lipid uptake and recycling rely heavily on neuron-glia interactions[3,27,28,59]. Based on these findings, we hypothesize that neuronal LpRs interact and acquire their lipid cargo from astrocyte-derived lipoproteins.

To validate and visualize the interactions between astrocytes and larval LNvs, we used the TRACT (TRAnsneuronal Control of Transcription) technique[60], where GFP expression is induced by close interactions between two groups of cells. Consistent with published results, we observed that a small number of astrocytes (2–3) have trans-synaptic interactions with LNvs[61] (Supplementary Fig. 6). In addition, we performed dual-labeling experiments with astrocytes labeled by Alrm-Gal4[62] driving UAS-CD8::GFP expression and LNvs labeled by Pdf-LexA driving LexAop-tdTomato expression (Fig. 6a)[41]. The results showed that LNv dendritic arbors are closely associated with the processes of astrocytes in the LON region, indicating physical proximity and potential functional connections between astrocytes and LNvs.

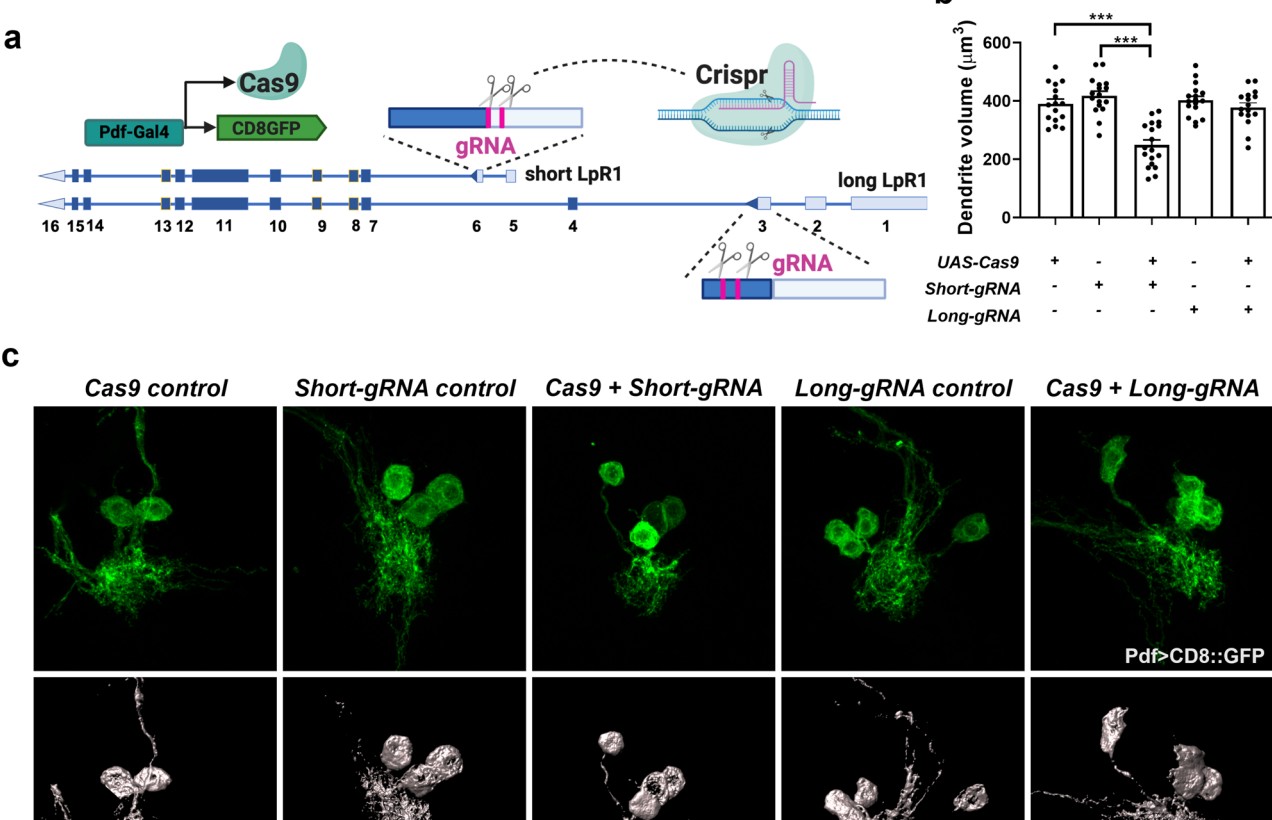

**Fig. 4 CRISPR/Cas9 mediated tissue-specific mutagenesis reveals the specific requirement of LpR1-short in neurons. a** Schematic diagram of CRISPR/Cas9-mediated tissue-specific mutagenesis to eliminate either the short or long isoforms of LpR1 in LNvs (created with BioRender.com). The Cas9 expression is driven by the LNv specific promotor (Pdf-Gal4>UAS-Cas9) and two sets of gRNAs (red bars) targeting either the long isoform-specific exon 3 or the short isoform-specific exon 6. **b** Quantifications of the reconstructed LNv dendrites (labeled by Pdf-Gal4>UAS-CD8::GFP) indicate that CRISPR/Cas9 mediated mutagenesis of LpR1-short produced a significant reduction of the LNv dendrite volume, while the long isoform-specific gRNA showed no difference from control groups. Data are presented as mean values +/− SEM. Statistical significance was assessed by one-way ANOVA with Tukey's post hoc test. ANOVA: $p < 0.0001$, $F = 19.39$, df = 79; *Cas9 control/Cas9 + Short-gRNA*: $p < 0.0001$; *Short-gRNA control/Cas9 + Short-gRNA*: $p < 0.0001$; *Cas9 control/Cas9 + Long-gRNA*: $p = 0.9840$; *Long-gRNA control/Cas9 + Long-gRNA*: $p = 0.8120$. $n = 16$, 17, 18, 17, and 16 for *Cas9 control, Short-gRNA control, Short-gRNA, Long-gRNA control* and *Long-gRNA*. *n* represents individual larval brain samples. ***$p < 0.001$. **c** Representative projected confocal images (green, top panels) and 3D reconstructions (gray, bottom panels) of LNv soma and dendritic regions are shown (repeated in at least 16 brains).

To identify lipoproteins released by astrocytes that participate in neuron-glia lipid shuttling, we first compared fly homologs of 125 human lipid transfer proteins against the larval astrocyte RNA-seq dataset[63,64] (Supplementary Data 1). Among the 87 fly homologs, 29 are present and 11 are enriched in larval astrocytes (Cutoff: expression value >40 and log2 fold change to brain expression level >1). These eleven glia-derived lipoproteins include five previously characterized genes, *GLaz, Neural Lazarillo (NLaz), Niemann-Pick type C-2b (Npc2b), prolonged-depolarization-afterpotential-is-not-apparent (pinta), Sterol-carrier-protein X-related thiolase (ScpX)*, and six genes with unknown functions (Supplementary Data 2). The candidates were then subjected to transgenic RNAi[65] screens using the *Drosophila* astrocyte-specific enhancer driver, Alrm-Gal4[62], followed by quantifications of LNv dendrite morphology (Fig. 6b, c).

Three transgenic RNAi lines, targeting either GLaz, a known glia-derived secreted protein, or CG12926 and CG30392, two uncharacterized proteins with lipid-binding domains, generated

significant reductions in LNv dendrite volume (Fig. 6c, d). The rest of the RNAi lines, as well as the ones targeting two major lipoproteins in *Drosophila, apolpp* and *Apoltp*, did not produce LNv dendrite phenotypes (Fig. 6c). Among these three candidate genes, we were particularly interested in GLaz, the fly homolog of human APOD, a small extracellular molecule with a lipocalin domain that binds to lipids. Human APOD is associated with aging and neural injury[66]. In *Drosophila*, GLaz regulates longevity and stress resistance in adult flies and was previously identified in the CNS glia[38,39]. However, its potential functions in neural development were not determined previously. In addition, recent studies in adult flies indicate that GLaz is involved in lipid transfer from neurons to glia[28], supporting GLaz as a strong candidate for our follow-up studies.

To validate the screen results, we performed additional RNAi knock-down experiments using a second RNAi line targeting *GLaz* and also observed a dendrite reduction phenotype (Fig. 6c). Importantly, *GLaz* knock-down in astrocytes also led to a

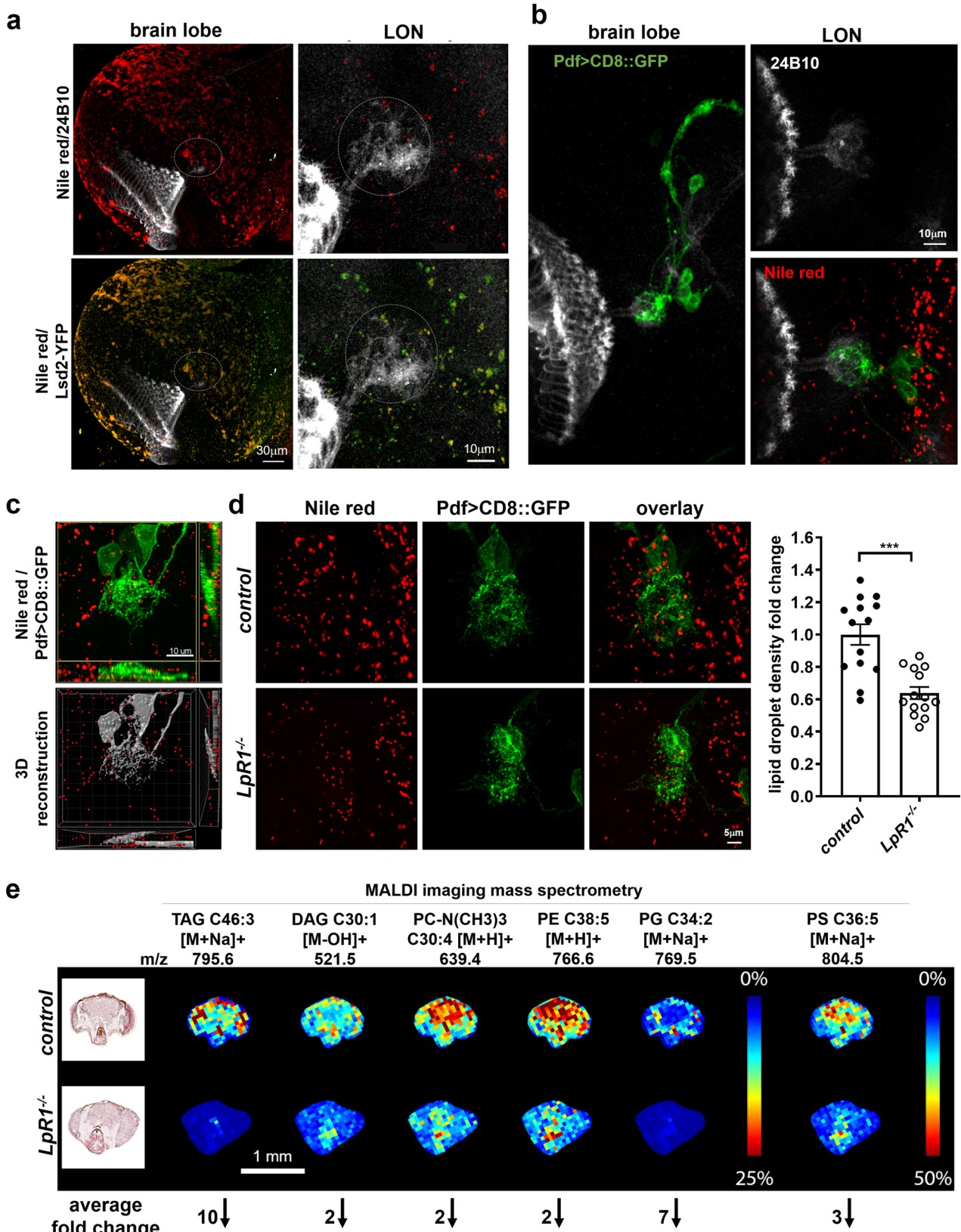

significant reduction in lipid droplet density in LON, similar to the phenotype we observed in *LpR1* mutants (Fig. 6e, f). Together, our genetic studies show that astrocyte-derived GLaz is involved in regulating LNv dendrite growth and brain lipid droplet accumulation, supporting its potential role as a major apolipoprotein mediating lipid trafficking in the brain.

**GLaz interacts with LpR1-short in the larval brain.** Our studies demonstrated that both neuronal expression of LpR1-short and astrocyte expression of GLaz are required for the normal development of LNv dendrites and brain lipid homeostasis. Next, we sought to examine the physical and functional interactions between these two proteins.

**Fig. 5 LpR1 regulates lipid homeostasis in the larval brain. a** Lipid droplets are detected in the larval optic neuropil (LON). Nile red staining (red) and Lsd2-YFP (green) expression both label lipid droplets in the larval brain. Representative projected confocal images of the larval brain lobe (left) and the LON region (circled region, right) are shown (observed in at least 10 brains). **b** Lipid droplets (red) are found in the vicinity of the LNv dendrites. Representative confocal images show the 24B10 stained axonal terminals of larval photoreceptor cells (gray) making contacts with LNv dendrites (labeled by CD8::GFP, green) within LON (observed in at least 10 brains). **c** The density of lipid droplets in LON is quantified by 3D reconstructions. Representative projected confocal images (top: green, LNv dendrites; red, lipid droplets) and 3D reconstructions for lipid droplet quantification (bottom: gray, LNv dendrite; red, lipid droplets) are shown (observed in at least 10 brains). **d** Compared to controls, $LpR1^{-/-}$ mutants shows decreased lipid droplet density in LON. Representative confocal images and quantifications are shown. Data are presented as mean values $+/-$ SEM. Statistical significance was assessed by a two-tailed Student's $t$-test. $p < 0.0001$, $t = 4.879$, df $= 26$. $n = 14$ for both groups. $n$ represents individual larval brain samples. ***$p < 0.001$. **e** Representative images from MALDI mass spectrometry analysis revealed a general reduction of lipid contents in the $LpR1^{-/-}$ the mutant brain. Representative H&E stained adult fly brain sections (left) and the MALDI scan images (right) are shown (observed in at least 4 biological repeats). The lipid species, their corresponding $m/z$ spectrum, and the scale of the heatmap are as indicated. Bottom: The average fold of reduction in the $LpR1^{-/-}$ mutant, as compared to wild-type controls, are shown as numbers next to the arrow.

We performed genetic analyses using a loss-of-function mutant of *GLaz*, *GLazΔ2*, which contains a null mutation and does not produce *GLaz* transcripts[38]. Deficiencies in either *GLaz* or *LpR1* generate strong dendrite reduction phenotypes in LNvs. Compared to wild-type controls, both *GLaz* and *LpR1* heterozygous mutants $GLaz^{\Delta2/+}$ and $LpR1^{+/-}$ showed mild but statistically significant reductions in their dendrite volume, while the trans-heterozygotes of two mutant alleles ($GLaz^{\Delta2/+}$; $LpR1^{+/-}$) displayed an enhanced phenotype (Fig. 7a), suggesting that *GLaz* and *LpR1* function in the same pathway in regulating LNv dendrite development[38].

To determine the distribution of GLaz protein in the larval brain, we first examined a transgenic line expressing a GLaz protein with a C-terminus GFP tag and is driven by a 1.8 Kb GLaz enhancer (GLaz > GLaz-GFP) (gifted by Dr. Maria D. Ganfornina) (Fig. 7b). Previous studies demonstrated that the expression of GLaz > GLaz-GFP rescues the loss-of-function mutant and potentially serves as an indicator for the endogenous level and distribution of the GLaz protein[38,67]. In the 3rd instar larval brain, we observed widely distributed GLaz > GLaz-GFP signals that form small puncta concentrated in several neuropil regions, including the dorsal protocerebrum and optic lobes (Fig. 7b, top).

To test whether GLaz>GLaz-GFP expression reflects the endogenous localization of the GLaz protein, which is known to be a secreted protein produced by glia[38,67], we examined the effects of blocking astrocyte secretion using a dominant-negative form of Rab1 (Rab1-DN) driven by the Alrm-Gal4[46]. This manipulation blocks the secretory pathway in astrocytes by affecting membrane trafficking between the ER and Golgi and drastically affected the GLaz > GLaz-GFP localization. In contrast to the diffuse distribution pattern seen in the control group, GLaz>GLaz-GFP signals here are largely concentrated in the astrocyte soma (Fig. 7b, bottom). By comparing the GLaz > GLaz-GFP distribution pattern between a control group and one in which secretion from astrocytes is blocked, we concluded that GLaz-GFP is secreted by astrocytes, consistent with the expected localization of the endogenous GLaz protein.

Notably, we found numerous GLaz-GFP puncta localized near or on the surface of LNv dendritic arbors in the larval LON, while few GFP puncta were observed within the neuron (Fig. 7c, Supplementary Fig. 7). By 3D reconstruction, we quantified the GLaz-GFP puncta localized within a 1 μm radius of the LNv dendrites and identified 51.3 $+/-$ 13.8 puncta per brain lobe (mean $+/-$ STDEV, $n = 9$). At the same time, only one GLaz-GFP punctum was found within the LNv soma in the nine brain samples (Supplementary Fig. 7). This observation suggests that GLaz-GFP performs lipid transfer at the surface of LNv dendritic arbors.

We also evaluated the native expression pattern of GLaz protein using a protein trap line generated by the Recombination-Mediated Cassette Exchange (RMCE) technique. The $GLaz^{MI02243\text{-}GFSTF.0}$ line expresses a GFP-tagged GLaz protein (GLaz-MiMIC-GFP) from its endogenous locus (Supplementary Fig. 8a)[43,44]. In the larval CNS, GLaz-MiMIC-GFP showed a diffuse distribution and was observed near the surface of LNv dendrites, similar to the results obtained through GLaz > GLaz-GFP (Fig. 7b, c, Supplementary Figs. 7, 8), albeit with a less punctuated appearance. In addition, blocking secretion in astrocytes through the expression of a dominant-negative Rab1 also led to the accumulation of GLaz-MiMIC-GFP in the astrocyte soma (Supplementary Fig. 8b).

Using two different GFP-tagged GLaz proteins, we confirmed that GLaz is secreted by astrocytes and is localized close to the surface of the LNv dendrite (Fig. 7b, c, Supplementary Figs. 7, 8). Although additional manipulations and in vivo imaging experiments are needed to reveal how GLaz is trafficked within the synaptic region, our results suggest that astrocyte-derived GLaz is likely recruited by neuronal lipoprotein receptors onto the surface of LNv dendrites, where it delivers lipid cargo without being internalized (Fig. 7c, Supplementary Figs. 7, 8c).

Next, we examined the biochemical properties of the *Drosophila* GLaz protein using an HA-tagged transgene driven by the astrocyte enhancer Alrm-Gal4 (Fig. 7d)[68]. Consistent with previous publications, we detected two bands around 27–30 kDa with anti-HA antibody, suggesting that GLaz-HA is correctly expressed and modified by glycosylation, similar to the modification reported in human APOD (Supplementary Fig. 9a)[69]. Notably, when concentrated during affinity purification by the anti-HA beads, GLaz-HA forms tetramers and dimers that can be destabilized into monomers by heat treatment or adding the reducing agent DTT. Therefore, GLaz-HA forms polymers when concentrated, another property that was recently reported for human APOD protein[70] (Supplementary Fig. 9a).

To determine whether GLaz directly binds to the LpR1 receptor, different *Drosophila* stocks, Alrm-Gal4>UAS-GLaz-HA and elav-Gal4>UAS-LpR1-short/long-GFP were generated for the co-immunoprecipitation (Co-IP) experiments. HA-tagged GLaz protein was extracted from 30 larval brains and loaded onto anti-HA magnetic beads. These beads were subsequently used to pulldown protein extracts containing LpR1-short/long-GFP. 10% of input protein extracts from each genotype were included as positive controls. Both LpR1-short-GFP and LpR1-long-GFP are readily detected in the larval brain extracts by western blots using the anti-GFP antibody (Fig. 7d, left panel). Importantly, only LpR1-short-GFP, but not the LpR1-long-GFP, binds to GLaz-HA and is pulled down in this set of experiments (Fig. 7d, right panel), suggesting that GLaz binds specifically to the short isoform of LpR1.

To confirm the direct interaction between LpR1-short and GLaz, we performed the reverse version of the co-IP experiments

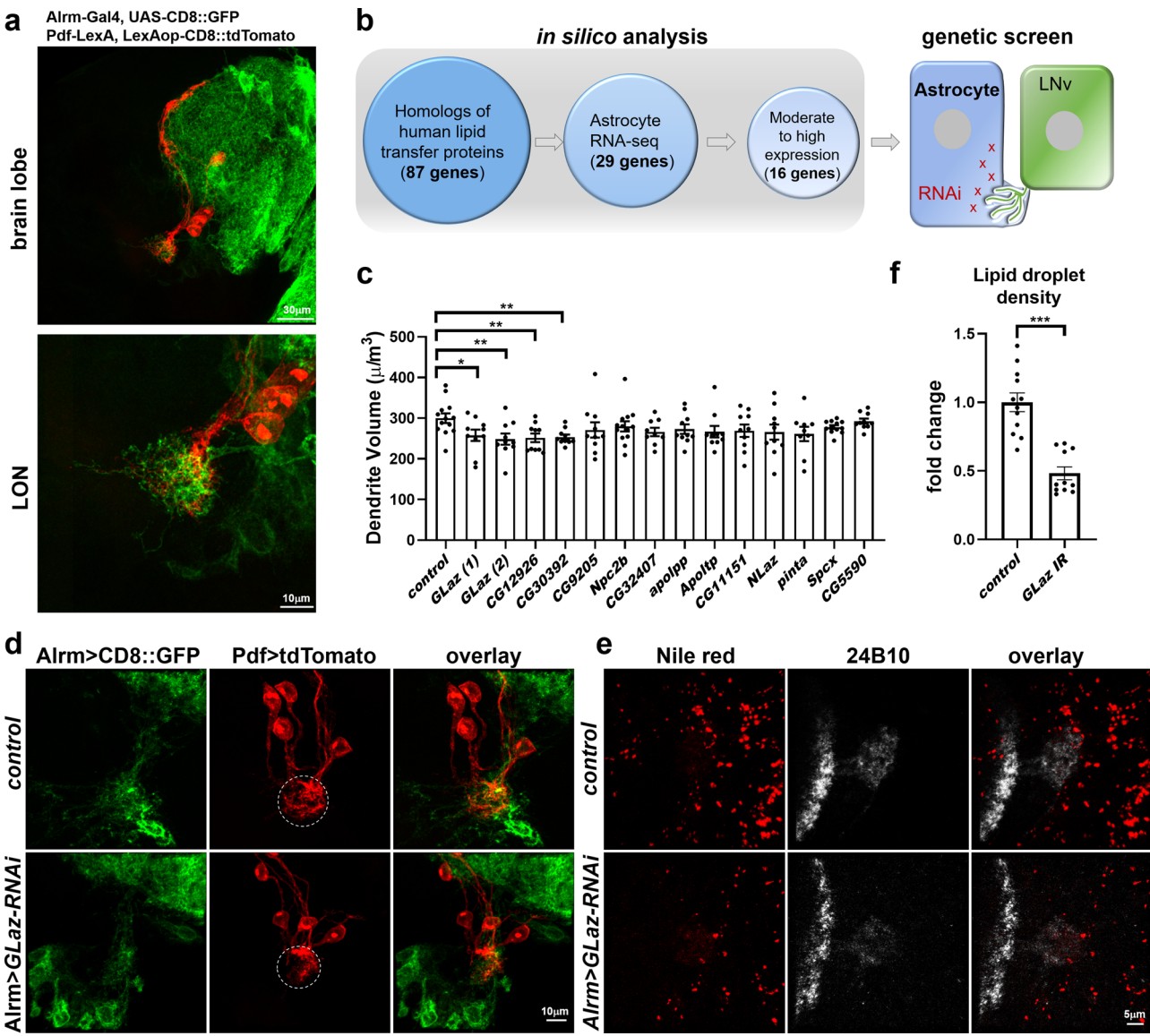

**Fig. 6 Genetic screens reveal the function of astrocyte-derived GLaz in dendrite development and lipid homeostasis. a** Astrocyte processes (labeled by Alrm-Gal4 driving CD8::GFP, green) and LNv dendrites (labeled by Pdf-lexA driving LexAop-tdTomato, red) show close interactions with the LON (observed in at least 10 brains). **b** A schematic representation of analyses performed to identify potential astrocyte-derived ligands for LpR1-short. Candidate lipoproteins were found by combined bioinformatic analyses and genetic screens of fly homologs of human lipid transfer proteins enriched in astrocytes. Genetic screens were performed using astrocyte-specific RNAi knock-down of candidate genes and subsequent quantification of LNv dendrite volume. **c** Quantifications of LNv dendrite volume. Transgenic RNAi knock-down of lipid transport proteins GLaz, CG12926, and CG30392 in astrocytes leads to significant LNv dendrite volume reduction. Data are presented as mean values $+/-$ SEM. Statistical significance was assessed by a two-tailed Student's $t$-test comparing with the control sample. *GLaz (1)*: $p = 0.0297$, $t = 2.333$, $df = 21$; *GLaz (2)*: $p = 0.0097$, $t = 2.846$, $df = 21$; *CG12926*: $p = 0.0075$, $t = 2.956$, $df = 21$; *CG30392*: $p = 0.0044$, $t = 3.194$, $df = 21$; *CG9205*: $p = 0.1874$, $t = 1.363$, $df = 21$; *Npc2b*: $p = 0.2483$, $t = 1.183$, $df = 24$; *CG32407*: $p = 0.0550$, $t = 2.038$, $df = 20$; *apolpp*: $p = 0.1263$, $t = 1.592$, $df = 21$; *Apoltp*: $p = 0.0831$, $t = 1.820$, $df = 21$; *CG11151*: $p = 0.1164$, $t = 1.638$, $df = 21$; *NLaz*: $p = 0.1178$, $t = 1.631$, $df = 21$; *pinta*: $p = 0.0721$, $t = 1.899$, $df = 20$; *Spcx*: $p = 0.1057$, $t = 1.684$, $df = 23$; *CG5590*: $p = 0.6180$, $t = 0.5062$, $df = 21$. $n = 13, 10, 10,$ 10, 10, 10, 13, 9, 10, 10, 10, 10, 9, 12, and 10 for *control, GLaz (1), GLaz (2), CG12926, CG30392, CG9205, Npc2b, CG32407, apolpp, Apoltp, CG11151, NLaz, pinta, Spcx,* and *CG5590. n* represents individual larval brain samples. *$*p < 0.05$, $**p < 0.01$. **d** Representative projected confocal images showing the reduced LNv dendrite volume (red) by knocking down GLaz in astrocytes (green) (repeated in at least 10 brains). **e–f** Astrocyte-derived GLaz regulates lipid droplet density in LON. Representative projected confocal images (**e**) and quantifications (**f**) showing that lipid droplet density in LON is reduced when GLaz expression is knocked down in astrocytes. Data in **f** are presented as mean values $+/-$ SEM. Statistical significance was assessed by a two-tailed Student's $t$-test. $p < 0.0001$, $t = 6.178$, $df = 21$. $n = 12$ and 11 for *control* and *GLaz IR*. n represents individual larval brain samples. $***p < 0.001$.

using anti-GFP antibody-conjugated magnetic beads to pull down LpR1-short-GFP or LpR1-long-GFP, followed by incubation with the brain extract containing GLaz-HA. Consistently, this set of co-IP experiments showed that GLaz-HA is pulled down by, and thus binds specifically to, LpR1-short-GFP, but not LpR1-long-GFP (Supplementary Fig. 9b).

Taken together, our data clearly demonstrated genetic and physical interactions between GLaz and LpR1-short, suggesting that GLaz is a specific astrocyte-derived ligand for the neuronal lipoprotein receptor LpR1-short. While the exact lipid cargo(s) of GLaz remains unknown, it is likely that LpR1-short captures GLaz on the surface of a neuron or its processes for lipid

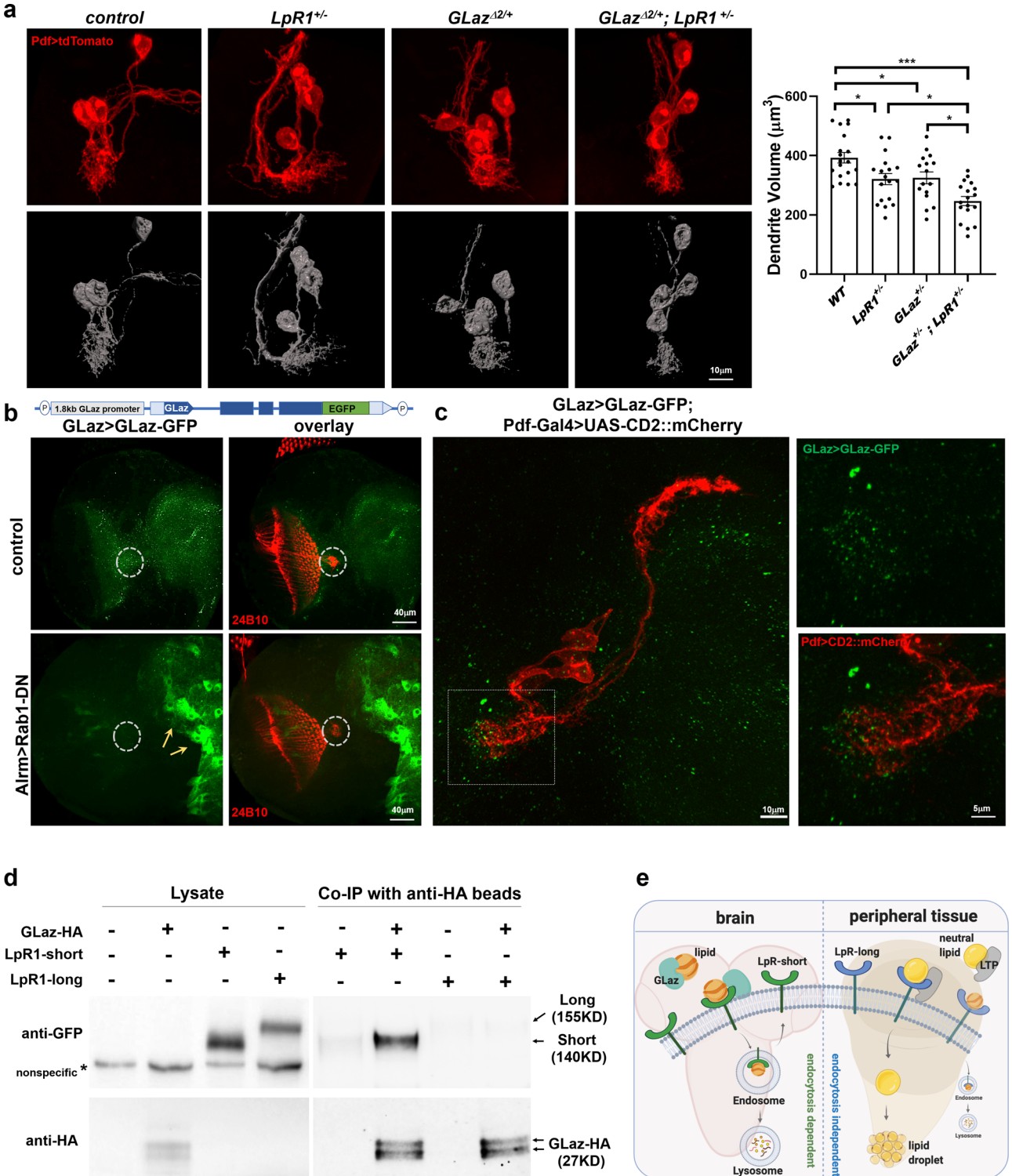

recruitment, then mediates endocytosis-dependent lipid uptake to support both the development and function of the neuron, as well as lipid homeostasis in the brain. This pathway is distinct from the one identified in the fly peripheral tissues, where LpR1-long interacts with LTP and mediates endocytosis-independent lipid uptake (Fig. 7e).

## Discussion

Lipid-mediated communication between glia and neurons is essential for brain lipid homeostasis and serves critical functions in neural development and synaptic function[10,26,29]. Here, using the developing *Drosophila* larval brain as a model, we investigate how neurons acquire their lipid supply from neighboring astrocytes and the regulatory mechanisms associated with the neuron–glia lipid trafficking. Our genetic and RNA-seq analyses reveal that lipid uptake in fly neurons is mediated largely by short isoforms of the LpR1 receptor, which recruits a lipid complex through direct interactions with astrocyte-derived apolipoprotein GLaz/APOD. This study identifies specific molecular carriers mediating neuron-glia lipid shuttling and reveals the isoform-

**Fig. 7 GLaz has functional and physical interactions with LpR1-short. a** *GLaz* and *LpR1-short* show genetic interactions in supporting LNv dendrite development. Left: representative projected confocal images (red) and 3D reconstructions (gray) of the LNv. Right: the quantifications of LNv dendrite volume. Compared to wildtype controls, the trans-heterozygous mutants (*GLaz$^{+/-}$*; *LpR1$^{+/-}$*) display a large reduction in the LNv dendrite volume, significantly lower than controls and both heterozygous *LpR1* and *GLaz* mutants. Data are presented as mean values $+/-$ SEM. Statistical significance was assessed by one-way ANOVA with Tukey's post hoc test. ANOVA: $p < 0.0001$, $F = 11.53$, df = 67; WT/$LpR1^{+/-}$: $p = 0.0257$; WT/$GLaz^{+/-}$: $p = 0.0486$; WT/$LpR1^{+/-}$; $GLaz^{+/-}$: $p < 0.0001$; $LpR1^{+/-}$/$LpR1^{+/-}$;$GLaz^{+/-}$: $p = 0.0226$; $GLaz^{+/-}$/$LpR1^{+/-}$;$GLaz^{+/-}$: $p = 0.0187$. $n = 19$, 18, 16, and 18 for WT, $LpR1^{-/-}$, $GLaz^{+/-}$, and $LpR1^{+/-}$; $GLaz^{+/-}$. $n$ represents individual larval brain samples. *$p < 0.05$, ***$p < 0.001$. **b** GLaz enhancer-driven GFP-tagged GLaz (GLaz > GLaz-GFP) is secreted from astrocytes. Top: a schematic diagram illustrating the elements included in the GLaz>GLaz-GFP transgene. Bottom: Representative confocal images show that blocking astrocytes' secretion by expressing a dominant-negative form of Rab1 (Rab1-DN) leads to the accumulation of GLaz>GLaz-GFP signals (green) in somas of astrocytes (arrows), in contrast to a diffused pattern observed in controls (observed in at least 10 brains). 24B10 staining (red) labels the LON (dashed circles). **c** GLaz > GLaz-GFP puncta localize near or on the surface of the LNv dendrites (observed in at least 9 brains). Left: Representative projected confocal images demonstrate the broad distribution of GLaz>GLaz-GFP (green) in the third instar larval brain. Right: Zoomed-in images of the LNv dendrites (Pdf > CD2::mCherry, red) show GLaz-GFP puncta in close proximity with LNv dendrites. **d** Co-IP experiment shows that GLaz binds to LpR1-short, but not LpR1-long. Both GFP-tagged LpR1-short (~140 KD) and LpR1-long (~150 KD) are expressed in neurons and detected by the anti-GFP antibody on the western blot. But only LpR1-short-GFP was found to be associated with GLaz-HA in the co-IP experiments using anti-HA beads. GLaz-HA is detected as two bands around 27KD using the anti-HA antibody. Asterisk (*) indicates a nonspecific band detected by anti-GFP antibody in the larval brain lysate. One representative result from two independent repeats is shown. **e** A proposed model for the distinct lipid recruitment in the *Drosophila* brain and peripheral tissues mediated by short vs. long isoforms of LpR1 (created with BioRender.com). LpR1-long mediates neutral lipid uptake and lipid droplet accumulation in peripheral tissues in an endocytosis-independent way, facilitated by its interactions with LTP. In contrast, LpR-short is the major isoform expressed in the fly CNS and likely recruits lipids through an endocytosis-dependent pathway, facilitated by its interactions with the glia-derived apolipoprotein GLaz.

specific expression of lipoprotein receptors as a mechanism that determines cell-type-specific recruitment of distinct lipid cargos.

**Functional diversity of lipoprotein receptors generated by isoform-specific regulation.** Exon mapping of cell-specific RNA-seq libraries revealed that only short isoforms of *LpR1* are expressed in LNvs and are upregulated by chronic elevation of neuronal activity (Fig. 1)[36]. Expanding upon those studies, we examined additional tissue-specific RNA-seq datasets and performed qFISH analyses (Fig. 2), which strengthened the conclusion that the short-isoforms of *LpR1* are CNS-specific and are predominately expressed in neurons, while the long-isoforms of LpR1 are mainly expressed in peripheral tissues and mediate endocytosis-independent lipid uptake (Fig. 7e)[33,34]. In addition, our genetic and biochemical analyses further reveal the functional significance of isoform specificity, including its impact on the receptor's distribution, binding partners, and ability to support specific types of lipid trafficking. These distinctions highlight transcriptional regulation as a key mechanism controlling the cell-type-specific distribution of lipoprotein receptors, their lipid cargos, and uptake efficacy. Results obtained from this study clearly indicate that the molecular and functional diversity of lipoprotein receptors far exceeds our current understandings, which are mostly based on DNA sequence analyses. Single-cell transcriptome analyses with sufficient resolution to identify isoform-specific splicing events could potentially reveal the capacity and specificity of the lipid uptake machinery within each individual cell type, helping us better understand the regulatory mechanisms underlying the cell-type-specific lipid recruitment.

**Bidirectional neuron-glia lipid shuttling supports neural development and plasticity.** Lipid shuttling between neurons and glia contributes to energy balance, neural protection, synapse development, and function, and potentially utilizes conserved molecular machinery that is shared among different organisms. Studies in *Drosophila* and murine neuron-glia coculture systems have demonstrated that neuronal activity alters the metabolic programs of both neurons and glia, leading to the transfer of neuronal lipids into glia for detoxification and storage in the form of lipid droplets. Notably, in both systems, vertebrate ApoE is able

to function as the lipid carrier supporting neuronal lipid transfer and lipid droplet accumulation in glia[28,31].

On the other hand, in the developing larval CNS, we found an increased lipid demand in neurons coping with chronically elevated input activity. This increase is likely met, at least partially, by enhancing neuronal lipid uptake through the activity-induced upregulation of *LpR1* expression. These observations demonstrate a strong influence imposed by activity on neurons' capacity for lipid recruitment through its effects on lipoprotein receptors[36]. Together with earlier studies, our findings suggest that both sides of neuron-glia lipid shuttling are regulated by neuronal activity, and the regulatory mechanisms controlling the expression level of lipoprotein receptors and their interactions with specific ligand molecules likely have functional significance in activity-dependent structural and functional plasticity in the nervous system.

Recent studies in adult *Drosophila* compound eyes have demonstrated that GLaz is involved in lipid transfer from neuron to glia[28], while our studies illustrated the interaction between GLaz and LpR1 and a possible role for GLaz in delivering lipid to neurons (Fig. 7e). Therefore, GLaz appears to be involved in both sides of the lipid trafficking and potentially serves as a key molecular target for regulatory mechanisms controlling lipid homeostasis in the fly brain.

**Lipid cargos delivered to neurons by GLaz/ApoD.** The basic structure, molecular composition, and developmental processes of a synapse are shared among vertebrate and invertebrate systems. While synaptogenesis in mammalian neurons relies on cholesterol production by glial cells and its delivery by ApoE-containing lipoprotein complexes[10], whether the cholesterol and/or ApoE-like lipid carriers are required for building synapses in *Drosophila* neurons is not known. Importantly, the *Drosophila* genome does not contain a homolog of ApoE. It is also reported that flies do not have the ability to synthesize cholesterol, and only obtain it through their diet to produce essential hormones[71]. The contrast between these two systems suggests exciting possibilities for us to study the lipids involved in synapse construction by understanding the differences and similarities between lipid recruitment in *Drosophila* vs. mammalian neurons.

Our study demonstrates that the short isoforms of LpR1 recruit lipids and support dendrite morphogenesis through their

interactions with the astrocyte-secreted lipoprotein GLaz, the homolog of human APOD. Given the strong dendrite development phenotype, as well as the reduced life span and stress resistance observed in GLaz loss-of-function mutants, GLaz's hypothesized lipid cargo is likely a critical contributor towards synapse development and neuronal functions in the fly CNS. Currently, only a limited number of putative lipid ligands have been identified for GLaz's homologs; grasshopper *Lazarillo* binds to retinoic acid and fatty acids[72], and human APOD binds to arachidonic acid (AA) and progesterone (PG)[73,74]. Whether these lipids or other types of hydrophobic ligands bind to *Drosophila* GLaz has not been studied. Similar to human APOD, we observed dimeric and tetrameric GLaz in the larval brain extract (Supplementary Fig. 9a). This suggests that, instead of only being able to accommodate a small hydrophobic ligand as a single unit, the GLaz protein could participate in different types of lipoprotein complexes and exhibit distinctive behaviors in vivo through its multimeric forms[70].

APOD's function in the CNS has long been underestimated, despite the fact that APOD is highly elevated during aging and neural injury[66]. When examining recent human and mouse astrocyte RNA-seq data, we found that, although ApoE is the highest expressing apolipoprotein in mouse astrocytes, the most abundant apolipoprotein expressed in human astrocytes is APOD, strongly suggesting its functional importance in the CNS (Supplementary Fig. 10)[75]. In both *Drosophila* and mammalian systems, GLaz/APOD are produced by astrocytes and have both anti-oxidation and anti-inflammatory protective functions[23,66,67,76]. Given the similarities between GLaz's and APOD's functional and biochemical properties, our findings on the GLaz-LpR1 interaction in *Drosophila* may facilitate the identification of a mammalian lipoprotein receptor that interacts with APOD and provide new insights into its protective role in aging and neurodegenerative disorders.

## Methods

**Fly strains**. The following lines were used (in the order of appearance in figures): (1) Pdf-Gal4, Bloomington Stock Center (BDSC): 6899, 6990, (2) UAS-CD8::GFP, BDSC: FBst0005137, (3) LpR1^MI14131-TG4.1, BDSC: 67514, (4) Pdf-LexA, LexAop-CD8::tdTomato, (5) UAS-RedStinger, BDSC: 8547, (6) UAS-LpR1G-GFP, (7) UAS-LpR1H-GFP, (8) UAS-mCherry, (9) UAS-Rab5-tdTomato, (10) UAS-Lsd2-YFP, Kyoto Stock Center: CPTI-001655, (10) LpR1^-/- (Df(3R) lpr1), BDSC: 44236, (11) Canton-S, BDSC: 64349, (12) Alrm-Gal4, BDSC: 67031, (13) RNAi lines: LpR1^RNAi, BDSC: 27249; GLaz^RNAi, BDSC: 67228; GLaz^RNAi, Vienna *Drosophila* Resource Center (VDRC): v107433; NLaz^RNAi, VDRC: v101321; Npc2b^RNAi, BDSC: 38238; pinta^RNAi, VDRC: v28263; ScpX^RNAi, BDSC: 51449; apolpp^RNAi, BDSC: 28946; Apoltp^RNAi, BDSC: 51937; CG11151^RNAi, BDSC: 51413; CG12926^RNAi, VDRC: v109580; CG32407^RNAi, VDRC: v103584; CG32407^RNAi, VDRC: v100022; CG5590^RNAi, BDSC: 66929; CG9205^RNAi, BDSC: 42588, (14) GLaz>GLaz-GFP (gifted by Dr. Maria D. Ganfornina), (15) UAS-YFP-Rab1S25N, BDSC: 9757, (16) GLaz^−/−, BDSC: 76610, (17) UAS-GLaz-HA, FlyORF: F003550, (18) Elav-Gal4, BDSC: 458, (19) UAS-Dicer2, BDSC: 24651, (20) Alrm>SNTG4, 5XUAS-CD4::tdGFP; Pdf-LexA, LexAop-mCD19 (gifted by Carlos Lois), (21) UAS-GCaMP7f, BDSC: 86320, (22) UAS-Cas9, BDSC: 54595, (23) Act5C-Cas9 lig4, BDSC: 58492, (24) pAC-U63>gRNA-gRNA-LpR1-short, (25) pAC-U63 > tgRNA-gRNA-LpR1-long, (26) GLaz^MI02243-GFSTF.0, BDSC: 60526.

**Fly culture**. Fly stocks are maintained in the standard cornmeal-based fly food in a 25 °C incubator with humidity control. For developmentally staged larvae, embryos with the desired genotypes were collected on plates containing grape juice agar supplemented with yeast paste. 1st instar larvae hatched within a 2 h time window. Larvae are cultured in either the light: dark (LD) condition with a 12 h light: 12 h dark light schedule; or in the constant light (LL) condition under 24 h light. Unless otherwise noted, all larvae were collected between ZT1-ZT3 (ZT: zeitgeber time in a 12:12 h light-dark cycle; lights-on at ZT0, lights-off at ZT12).

**RNA-seq data analysis**. RNA-seq data for LNv-LD and LNv-LL are from our previous publication (GEO: GSE106930). The *Drosophila* 3rd instar larval central nervous system (CNS) RNA-seq data were downloaded from ModEncode (ID-4658). Exons mapped to LpR1 gene were visualized using the integrated genome browser (IGV 2.7.2) in Fig. 1b. In Fig. 2a, RNA-seq data for the CNS, digestive

system, fat body, imaginal disc, and salivary gland are from NCBI (GSM838792, GSM838789, GSM838784, GSM838752, GSM838787); and LNv, LNd, DN1, TH-specific data are from Abuzzi et al.[44] (GSM2052238, GSM2052220, GSM2052244, GSM2052262). The data from Fig. 2a were visualized with NCBI Genome Data Viewer. RNA-seq data for ApoD and ApoE expression in mouse and human astrocytes is from (GEO: GSE52564)[75].

**Quantitative fluorescence in situ hybridization**. Quantitative fluorescence in situ hybridization (qFISH) was performed using an RNAscope Fluorescent Multiplex Kit (Advanced Cell Diagnostics, 320850). Customized C1 DNA oligonucleotide probes were designed by Advanced Cell Diagnostics for both LpR1 short isoforms (Cat: 560661) and long isoforms (Cat: 560671). FISH experiments were performed on acutely dissociated larval brain cells[36]. Third instar larvae brains were dissected and transferred to a clean dish containing cold DPBS and were cut into smaller pieces by needles. After proteinase treatment (Collagenase/ Dispase [1 mg/ml] and liberase I [0.1 Wünsch units/ml] for 40 min at 25 °C), media neutralization, and centrifuge, cells were resuspended in 30 μl Schneider's Insect Medium and transferred onto chambered cell culture slides (VWR, 53106-306). The chamber slides were pretreated with 0.25 mg/ml Concanavalin A (Sigma). There are four LNvs in each brain lobe and about 20% of the dissociated LNvs were recovered. After 10 min at room temperature, the extra solution was removed from the slides. Adhered cells were then fixed in 4% paraformaldehyde in PBS for 10 min and then washed with PBST twice. Subsequent FISH steps were performed following the manufacture's protocol for adherent cells. LNvs were identified by either gene-specific enhancer driving GFP expression or by immunostaining using the anti-PDF antibody. Images are taken with a Zeiss LSM800 confocal microscope at NINDS Neurosciences Light Imaging Facility.

**Transgenic line generation**. LpR1H was amplified by primer pairs of "5- ACA-GATCTGCGGCCGCAATGGCCATAGA-3; 5-TGGCGGTACCCTCGAGA-TACGAGGATAT-3" using the LpR1H cDNA clone (DGRC, FBcl016099) as a template. LpR1G was amplified by combining the first 2 exons from LpR1D clone (DGRC, UFO10128) and the other exons from LpR1H clone. The first 2 exons of LpR1G were amplified by primer pairs of "5-ACAGATCTGCGGCCGCAT GGGGCGCATT-3" and "5-TTTGTTCGGTATGCAGTTG-3" using the LpR1D clone as a template. The remaining exons of LpR1G were amplified by the primer pair "5-AGGCCACTTGCTCGTCG-3" and "5-TGGCGGTACCCTCGAGA-TACGAGGATAT-3" using the LpR1H clone as a template. Then the full length of LpR1G was assembled using these two fragments and PCR reactions. LpR1G and LpR1H cDNAs were tagged with full-length GFP at the C terminus and inserted into pUAST plasmid to generate UAS-LpR1G-GFP and UAS-LpR1H-GFP constructs. Transgenic flies were generated using P-element transformation (Rainbow Transgenic Flies, Inc. CA, USA). Fly lines carrying insertions on the 2nd chromosome are mapped and balanced for genetic tests.

Generation of the Pdf>LpR1G-HA and Pdf>LpR1H-HA transgenic flies: LpR1H and LpR1G cDNA fragments were tagged with 3xHA sequence at the C terminus and inserted into the pDEST vector[77] containing a 1.6 Kb enhancer fragment cloned from the upstream region of the PDF gene[78]. The two constructs were then injected into PBac[y[+]-attp-3B]VK00037 to achieve site-specific integration of the transgenes into the 22A3 locus.

**Lipid droplet staining**. Larval brains were dissected at 120 h AEL and fixed in 4% of PFA/PBS at room temperature for 30 min, and washed in PBST (PBS, 3% TritonX) three times for 20 min each time. Fixed brains were incubated in the Nile Red solution (Sigma Aldrich, 19123, diluted 1:100 in PBST from a 100 mg/ml stock solution in acetone) at 4 °C overnight and then washed in PBST for 30 min each time. Larval brains were mounted on glass slides with the anti-fade mounting solution and imaged with a Zeiss LSM700 upright confocal microscope.

**Immunohistochemistry**. Larval brains collected at 120 h AEL were dissected and fixed in 4% PFA/PBS at room temperature for 30 min, followed by washing in PBST (0.3% Triton-X 100 in PBS), and incubating in the primary antibody overnight at 4 °C. On the next day, brains were washed with PBST and incubated in the secondary antibody at room temperature for 1 h before final washes in PBST and mounted on the slide with the anti-fade mounting solution. Primary antibodies used were mouse anti-PDF (DSHB PDF C7, 1:10), mouse anti-Chp (DSHB 24B10, 1:10) and rat anti-HA (Sigma, 11867423001, 1:200). Secondary antibodies (1:1000 dilution) used were goat anti-mouse Alex 488 (Invitrogen, A-32723), goat anti-mouse Alex 647 (Invitrogen, A-32728), donkey anti-mouse CY3 (Jackson Immuno Research Labs, 715165150), goat anti-rat Alex 647 (Invitrogen, A-21247). Images are taken either with either a Zeiss LSM700 confocal microscope in the lab or Zeiss LSM800 confocal microscope at NINDS Neurosciences Light Imaging Facility.

**Construction of the transgenic gRNA lines for CRISPR/Cas9 mediated mutagenesis**. Transgenic gRNA expression vectors were constructed using pAC-CR7T-gRNA2.1-nlsBFP[51], a gRNA cloning vector based on pAC (attB-CaSpeR4)[52]. Each of the final gRNA expression vectors contains two gRNA-expression cassettes in tandem and a nuclear BFP driven by Ubi-63E promoter[79]. The first gRNA-expression cassette contains CR7T Pol III promoter[80], while the

second contains U6:3 promoter[81]. Each gRNA targeting sequence was followed by a modified gRNA scaffold sequence gRNA2.1[82]. The targeting sequences for *LpR1* exon 3 (long isoform) are TGAAGTTGGCGCATGCGCAG and ATTATGTGGGCTAACTGCCG. The targeting sequences for *LpR1* exon 6 (short isoform) are GGAAAAAACTACGGAAAATG and TTTCGCGTTCTGGCTATAAA.

Efficiencies of these gRNA constructs were validated using the Cas9-LEThAL (Cas9-induced lethal effect through the absence of Lig4) assay[52]. Briefly, transgenic gRNA male flies are crossed to *Act5C-Cas9 lig4* females[53]. Because Act-Cas9 and the *lig4* null mutation are both on the X chromosome, the male progeny from the cross is *lig4* deficient and therefore cannot repair DNA double-stranded breaks (DSBs) through non-homologous end-joining. In contrast, the female progeny is heterozygous for *lig4* and thus can repair DSBs. High lethality in male progeny from this cross indicates high efficacy of the gRNAs. The lethality of the gRNAs was calculated as [1−(numbers of male progeny/numbers of female progeny)].

**MALDI TOF MS imaging.** The MALDI TOF MS imaging was carried out in MALDI MS Imaging Joint Facility at the Advanced Science Research Center of City University of New York-Graduate Center. High purity grade 2,5-dihydroxybenzoic acid (DHB), Phosphorus (red), and AgNO3 and trifluoroacetic acid (TFA) were purchased from Millipore Sigma-Aldrich (USA). Optima UHPLC/MS-grade methanol and water were purchased from Fisher Scientific (USA).

The fly brains were embedded in the OCT matrix and snap-frozen in Ethanol/dry ice slurry. The frozen tissue was cryosectioned at 12 μm thickness sections using CryoStar NX50 (Thermo Scientific, USA) at −15 °C set for both specimen head and the chamber. The tissue cryosections were then gently transferred onto the pre-cooled conductive side of the indium tin oxide (ITO)-coated glass slides (Bruker Daltonics, Bremen, Germany) for MALDI imaging. Mounted cryosections on ITO slides were desiccated in a vacuum for 45 min at room temperature, followed by matrix deposition using HXT M5 sprayer (HXT LLC., USA). Matrix DHB solution of 40 mg/mL in methanol/water (70/30, v/v) was deposited at a flow rate of 0.12 ml/min and a nozzle temperature of 85 °C for 10 passes. A spray velocity of 1300 mm/min, track spacing of 2 mm, N2 gas pressure of 10 psi, and flow rate of 3 L/min, and nozzle height of 40 mm were used.

For the detection of cholesterol, a solution of 8 mg/ml AgNO3, plus 0.1%TFA in methanol/water (75/25, v/v) was deposited at a flow rate of 0.05 ml/min and a nozzle temperature of 45 °C for 25 passes, with the tissue slides were heated up and maintained at 30 °C during the spray[83]. A spray velocity of 1000 mm/min, track spacing of 3 mm, N2 gas pressure of 10 psi and flow rate of 3 L/min, and nozzle height of 40 mm were used.

MALDI mass spectra were acquired in positive ion mode for DHB acquired by MALDI time-of-flight (TOF) mass spectrometer Autoflex (Bruker Daltonics, Germany). MS spectra were calibrated using red phosphorus as the standard for all experiments. The laser spot diameters were focused on a "Medium" modulated beam profile for 40 μm raster width. The imaging data for each array position were summed up by 500 shots at a laser repetition rate of 1000 Hz. Spectra were acquired in the mass range from $m/z$ 50 to 1000 with a low mass gate at 50 Da. Imaging data were recorded and processed using FlexImaging v3.0, and it was further analyzed using SCiLS (2015b). Ion images were generated with root-mean-square (RMS) normalization and a bin width of ±0.10 Da. The spectra were interpreted manually, and analyte assignment was achieved by referencing METLIN databases and previously published studies[54,84]. Cholesterol is detected at $m/z$ 493.2 as [cholesterol + Ag[107]]+ and at $m/z$ 495.2 as [cholesterol + Ag[109]]+.

The MALDI slides were retrieved immediately after the experiment, washed with 95% ethanol, and proceeded with standard H&E staining. The H&E images were acquired using Leica Aperio CS2 slide scanner and were used as the anatomical reference for mass spectra imaging. Only signals from the brain region are collected for quantification.

**TRAnsneuronal control of transcription (TRACT).** TRACT is a technique to monitor cell-cell contact in vivo[60,61]. Briefly, SNTG4 expression is driven by the alrm enhancer and is expressed in all astrocytes. Upon CD19 binding, SNTG4 is cleaved and releases the Gal4 into the nucleus of the receiver cell to activate the transcription of UAS dependent GFP. To analyze the LNv-astrocyte interactions, Pdf-LexA > LexAop-mCD19 was crossed with Alrm>SNTG4; 5XUAS-CD4::tdGFP. Transsynaptic interactions between LNv and astrocytes led to the induction of CD4::tdGFP expression in a small number of astrocytes.

**In vivo transgenic RNAi screen.** The Alrm-gal4, UAS-CD8::GFP; Pdf-LexA, LexAop-CD8::tdTomato line was crossed with transgenic RNAi lines targeting candidate molecules. Larvae brains were dissected at wandering late 3rd instar stage (120 h AEL). Brains were fixed with 4% PFA/PBS and followed by 3 rounds of wash in PBST. After mounting, at least 10 brains were imaged for the assessment of dendrite phenotype.

**Co-Immunoprecipitation.** To collect protein extracts for Co-IP experiments, we cultured the lines with the following genotypes: Alrm-Gal4>UAS-GLaz-HA, Elav-Gal4>UAS-LpR1G-GFP, and Elav-Gal4>UAS-LpR1H-GFP. Brains from 3rd instar larvae in each of these lines were dissected and homogenized with lysis buffer N-Per (Thermo Fisher, 87792) containing 0.1 mM DTT and proteinase inhibitor

cocktail (Sigma, P8340) or RNasin Plus RNase inhibitor (Promega, N2611). Lysate from 30 brains was incubated with anti-HA magnetic beads (Pierce, 88838) or GFP-Trap Magnetic agarose (Chromotek Biotechnology, gtmak-20) at 4 °C for 6 h to develop GLaz-HA or LpR1G/H-GFP on the beads. The lysate was removed after 6 hrs incubation. Then, lysate from LpR1G-GFP or LpR1H-GFP was incubated with GLaz incubated anti-HA beads and GLaz-HA lysate was incubated with 1G or 1H incubated anti-GFP beads at 4 °C overnight. Beads were washed 3 times with 0.5 mL wash buffer provided with the kit and a Protease inhibitor cocktail. Proteins were separated from the beads by incubating with protein sample buffer containing 500 mM DTT at room temperature for 1 h. Pulled-down proteins were subjected to precast Bis-Tris Protein Gel for Western Blot. 10% of input protein extracts (~3 brains) from each genotype were included as positive controls. After the proteins were transferred onto PVDF membrane, 5% of milk in TBST was used for blocking unspecific binding. The membrane was incubated with rabbit anti-GFP (Abcam, ab6556) or rat anti-HA (Sigma, 11867423001) primary antibody in a 1:2000 dilution at 4 °C overnight. After 3 rounds of TBST wash, HRP conjugated a secondary antibody (Thermo Fisher, SA1-200 and A18745) in a 1:10,000 dilution and was incubated for 1 h at room temperature. A chemiluminescence detection kit (Bio-Rad, 1705062) was used for membrane exposure.

**Quantification and statistical analysis.** For LNv dendrite volume quantification, images of LNvs were obtained from fixed whole-mount larval brains using a ×40 oil objective. Serial optic sections of 0.2–0.50 μm thickness were taken on a Zeiss-700 or Zeiss-880 Airyscan confocal microscope and processed by Imaris (×64 9.3.0) 3D image visualization software. The imaging settings were kept consistent in all experiments. For 3D reconstruction and quantification of LNv dendrites, the Surfaces module of the Imaris was used to detect the GFP intensity on dendritic arbors and reconstruct the surface that encapsulates the entire dendritic tree. The dendrite volume was exported from Imaris into Excel (16.45) and used for calculations.

For 3D Sholl analysis, the "Filament Sholl Analysis" Imaris XTension (published by Matthew Gastinger) was incorporated into Imaris. LNv dendrites were reconstructed in 3D by using the Filaments module of Imaris based on GFP intensity. The values for "Filament No. Sholl Intersections" were exported from Imaris into Excel and used for calculations.

For lipid droplet density quantification, images were processed by Imaris 3D image visualization software. The Spots module of the Imaris was used to detect the number of lipid droplets within a 3D volume of 52.1 μm × 52.1 μm frame with a thickness that covers the LNv dendrites or LON. The total number was exported from Imaris into Excel. The density was calculated by lipid droplets numbers divided by the brain volume of the quantified region.

For GLaz > GLaz-GFP puncta quantification, to calculate the GLaz-GFP puncta within the LNv soma, individual optic sections of the image stacks were viewed in the Slice mode and the GFP positive puncta within the LNv soma are manually counted. To examine the GLaz-GFP puncta in the close proximity of LNv dendrites, LNv dendrites are reconstructed using the Surface module, and the GLaz-GFP puncta are reconstructed by the Spots Module. Using the "Find Spot close to the Surface module" in Imaris XTension, and 1 μm as an arbitrarily set threshold, we calculated the GLaz-GFP puncta that are within the 1 μm radius of the LNv dendrite surface.

For the qFISH experiment, numbers of dots generated by the LpR1-short and LpR1-long probes in each LNv neuron were manually counted for quantifications.

Graphing and statistical analysis of the quantifications were performed using Graphpad Prism (9.0.0.) For statistical analyses between two groups of samples, a two-tailed unpaired student *t*-test was performed; for experiments with more than two groups, one-way ANOVA analysis followed by multiple comparisons Tukey post-hoc test was performed, with the exception of the astrocyte apolipoprotein screening presented in Fig. 6c, where the unpaired student *t*-tests were performed between the control group and the individual RNAi knock-down groups. The exact value of sample number n, the statistical tests used and the confidence intervals, and the precision measures for individual experiments are included in the figure legends. Most quantitative data are presented as bar plot overlaid with dot plot; bar plot shows the mean (height of the bar) and SEM (error bars); dot plot displays individual data points. *n* represents the number of dissociated LNvs or GFP positive neurons in Figs. 1d and 2c; n represents the quantified images in Fig. 2e; *n* represents the number of larvae brains in Figs. 3b-d, 4b, 5d, 6c, f, 7a, Supplementary Fig. 4; *n* represents the number of adult brains in Supplementary Fig. 5. Statistical significances were assigned as: *$p < 0.05$, **$p < 0.01$, ***$p < 0.001$, ns: not significant.

**Reporting summary.** Further information on research design is available in the Nature Research Reporting Summary linked to this article.

## Data availability
All data supporting the findings in this study are available from the corresponding author upon reasonable request. The source data underlying Figs. 1d, 2c, e, 3b–d, 4b, 5d, 6c, f, 7a, and Supplementary Figs. 4, 5, and 10 are provided as a Source Data file. The RNA-seq data are available in modEncode (ID-4658) or NCBI (GEO: GSE106930; GSM838792; GSM838789; GSM838784; GSM838752; GSM838787; GSM2052238; GSM2052220; GSM2052244; GSM2052262; GEO: GSE52564). Source data are provided with this paper.

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

## Acknowledgements
We thank Dr. Joaquim Culi, Dr. Maria D. Ganfornina, and Dr. Carlos Lois for the mutant and transgenic fly lines; and Dr. Yihong Ye, Mark Stopfer, and Benjamin White for helpful discussions; and Dr. Carolyn Smith at the NINDS Light Imaging Core for technical support; and Tiffany Zhong for the technical support of MALDI MS imaging. Yuki Chen and Kelly Veerasammy are supported by the Sloan Foundation CSURP program; Ye He and Rinat Abzalimov are supported by PSC-CUNY Research Award Program. Bei Wang and Chun Han are supported by NIH grant R01NS099125. This research is supported by the intramural research program of the National Institutes of Health. Project number 1ZIANS003137.

## Author contributions
J.Y. and Q.Y. designed the experiments. J.Y. performed RNA-seq bioinformatics analyses. J.Y., E.S., E.S.C, J.S., Y.C., M.G., J.L., J. R., C. S., and Q.Y. performed data collection and analyses. B.W and C.H performed gRNA design and construction. Y.X.C., K.V., T.C., R. A., and Y.H. performed MALDI-TOF MS experiment and analysis. J.Y. and Q.Y. wrote the manuscript.

## Funding

## Competing interests
The authors declare no competing interests.
