## [Peer Review File · Nature Communications]

REVIEWER COMMENTS

Reviewer #1 (Remarks to the Author):

Brain specific lipoprotein receptors

Yin et al analyze lipid shuttling in the *Drosophila* nervous system. They report that the lipocalin Glial Lazarillo is secreted by astrocytes and interacts with an Apolipoprotein D homolog expressed by neurons. The topic is certainly interesting but the present study is not based on solid data and in several cases, even ignores the state of the art. Instead, the authors mix data obtained for mammalian models and *Drosophila* in a not permitted manner. It starts in the introduction where the claim is made that astrocytes secrete apolipoproteins - none of the *Drosophila* references given shows any of such data. Strangely the work of Brankatschk (eLIFE, 2014) is not even mentioned in the paper. Brankatschk et al. report that LpR1/2 is expressed primarily in glial cells in the larval brain (Brankatschk M, Dunst S, Nemetschke L & Eaton S (2014). Delivery of circulating lipoproteins to specific neurons in the *Drosophila* brain regulates systemic insulin signaling. *Elife* 3, 1–19.).

Likewise, the work of McFerrin et al., 2017 is not mentioned - which would be important in the context of lipid droplet formation (page 2). The authors focus on LpR1 and the Apolipoprotein D (APOD) homolog encoded by the *Drosophila* gene GLaz. They claim (page 3) that in mammals APOD is the most abundant type of apolipoprotein secreted by glia. Looking up the reference and the excellent data base in the Barres lab indeed shows minor expression of APOD in human astrocytes (comparable to neurons and less than endothelial cells) but the most dramatic expression is in oligodendrocytes (almost 10x more). Why are the claim and the data so far apart?

The manuscript starts with a lengthy description of the genomic organization and expression analysis of the two isoforms of the *Drosophila* LpR1 gene (5 pages). This is not what I would have expected in a Nature communications paper. Moreover, several of the experiments are way below standard in the *Drosophila* field. Expression analysis is either in silico or in dissociated cells! The fly brain is so small that single cell resolution is easily possible even in FISH experiments. In fact, the CRISPR technology is well established in the field and insertions of small tags into the endogenous genes are easily feasible and must be done to demonstrate expression of the different proteins. Localization studies using overexpression constructs are highly questionable. Likewise, the notion that the two different LpR1 isoforms (short and long) show different localization and thus possibly different function calls for isoform specific mutants which are quite easy to generate in *Drosophila*.

Line 158 The MIMIC insertion used by the authors is not an enhancer trap line! The analysis of the expression pattern is not convincing.

Line 163 Figure 2d - what is shown here. Costaining with elav or Repo should have been done.

Line 192 claims that the results are consistent with previous findings in the *Drosophila* PNS but no references are given to support the claim.

Lines 229-244 Do the authors claim that lipid droplets are in neurons? How should this be explained mechanistically? The authors use elav-Gal4 to target neurons, nsyb-Gal4 would be the cleaner driver. The finding is that elav-Gal4 driven RNAi based suppression of LpR1 results in fewer lipid droplets.

Line 248 MALDI-TOF MS experiments allow to visualize lipids in adult brain sections. Yes, but this is not done, the Figure shows head sections! and the head is full of very lipid rich fat body which might smear across the section and ruin the MALDI-TOF MS data or might obscure the data due to poor spatial resolution. The observed 10 fold decrease in TAG-content could be due to the fat body and the consequences of disruption of the global lipid

metabolism, which would be expected in null mutants, I guess.

From the images, I would say just the phospholipids - and not all of them - are really different in the brain. Those are membrane lipids, suggesting more membrane or depending on the type of phospholipid a different membrane composition (but I am not an expert)- not a storage phenotype

Line 264/265 Previous studies indicate that, in both vertebrate and invertebrate CNS, neuronal lipid uptake and recycling rely heavily on neuron-glia interactions. None of the reference listed shows this for invertebrates, likewise, one sentence later, Ma et al., never showed that astrocyte derived apolipoprotein has a critical function in synapse formation and function.

The analysis of GLaz continues in the same manner. Localization is studied using an uncharacterized (and not even shown) construct where a GLaz enhancer drives a GFP-GLaz fusion. The endogenous gene must be tagged or antibodies should be generated. Again, I did not see any link to mutants. Lines 334-338: How can the authors claim that "GLaz-HA is expressed in astrocytes and modified by glycosylation". They used alm-Gal4 to express the protein in astrocytes.

In summary, I don't think that this paper reports anything concrete about lipid uptake into neurons. The authors do not show that the LDs are in neurons, neither can they deduce this from the MALDI imaging experiments. They ignore published data (Brankatschk,2014) and link references to claims that are not supported in these studies.

Reviewer #2 (Remarks to the Author):

The manuscript by Yin et al. reports that trafficking of lipids from glial cells to neurons is important for activity dependent dendrite growth in the Drosophila central nervous system. A novel contribution of this study is to identify mechanisms of trafficking of lipids from glia to neurons and to show that activity regulates lipid shuttling through upregulation of the lipophorin receptor LpR1. The authors make several important discoveries regarding lipid shuttling in the central nervous system, including the identity of a putative glial derived lipoprotein Glaz that can promote dendrite morphogenesis, and LpR1 as a receptor for the lipoprotein on LNv dendrites. The developmental role for Glaz is particularly significant because prior studies had hypothesized developmental roles, but this had not yet been demonstrated. The findings should provide a map for further studies in mammalian systems. Evidence for functional interactions between Glaz and LpR1 are compelling. However, some of the conclusions lack strong support, dampening my enthusiasm for the manuscript somewhat without clarification of these concerns.

Major comments:

Line 182-185. One major conclusion of the manuscript, that different isoforms of LpR1 are localized differently in neurons, is limited because it appears to derive from UAS-Gal4 overexpression. The details of this are hard to find in the results, but in the methods I could only find UAS lines described. It was also not clear whether the transgenes were inserted into the same location in the fly genome. If not, this would make it hard to compare localization directly because the transgenes could be expressed at vastly different levels. Because overexpression could affect protein localization, the conclusion that LpR1-short and LpR1-long display distinct cellular localization in LNvs has several caveats. Another limitation to localization and rescue experiments is that only a single UAS line is used for overexpression of each of the isoforms.

line 212-213 it is proposed that LpR1-short isoforms may rescue by recruiting specific types of lipids for establishing synaptic connections. There is no strong evidence that this is the case, especially given that long and short isoforms can fully rescue the phenotype. The difference being that long isoforms cause an ectopic dendrite growth. Furthermore, because rescue is obtained with both short and long isoforms, isoform identity seems not to be important for this aspect of morphogenesis. The rescue by both also challenges the notion that differential localization shown in figure 3 is important for morphogenesis.

in the same section the authors suggest (on lines 218-220) that isoform specific expression of LpR1-short ensures proper synaptic connectivity, but the only difference here is that LpR1-long leads to exuberant dendrites growing off in one direction. There is no evidence that this difference in morphology leads to different synaptic connectivity either on those exuberant dendrites or the normal ones.

Line 243, it was not obvious how neuronal expression of LpR1 contributes to lipid droplet storage, presumably in glia? Would be useful to have some clarification of this.

Minor points:

In the abstract the authors claim that lipid transport mediated by Glaz is bidirectional. What is the evidence for this?

In lines 103-105 the authors imply that they have identified the APOD receptor as LpR1. The authors have identified the receptor for the APOD homolog Glaz. Although the evidence for conservation cited in the discussion is promising, it is still not clear whether this is going to generalize.

Is the requirement for lipid shuttling dendrite specific or is there any requirement for axon morphogenesis? These data may not be available and it is not an essential new experiment, but if the authors have any relevant information on this it could be interesting to comment or speculate on.

In reporting the results on the expression of LpR1 the authors go through a very detailed and rigorous description of isoform specific expression then move on to a less specific Gal4 line. The line and the data are not necessarily problematic, but this contributes very little coming after the RNA-seq and qFISH data. Perhaps starting with the Gal4 data then moving to the finer analysis would be more compelling.

TRACT technique is not in wide use so would be informative to have more description of the technique either in methods or results. Also, it is not clear that controls were performed (presumably standard control would be lack of lexA driver).

Figure 6c. Was the RNAi screen dataset corrected for multiple comparisons?

Figure 7: Are these images confocal projections? If so, what is the location of Glaz-GFP relative to dendrites in 3D? Rab1DN results indicate a defect in trafficking in astrocytes, however the authors conclude based on these results alone that GLaz is secreted from astrocytes. What is their justification for this conclusion?

Line 378-379 of discussion – authors have not addressed LpR2, so why do they generalize the results to both LpRs?

Reviewer #3 (Remarks to the Author):

This is a very interesting manuscript that identifies two major players that regulate astrocyte to neuron lipid transport that appear to play important roles in dendrite development. The experiments are elegantly designed and utilize a number of over expression, knock down, and protein labeling techniques to describe cellular localization, and confirm the involvement of LpR1 and GLaz in astrocyte to neuron lipid transport. The images presented in this paper are excellent visual representations of the experimental results. Likewise the manuscript itself is well written and makes a convincing case for the involvement of these two proteins in lipid transport and dendrite development.

There are several relatively minor deficiencies that I believe would improve the manuscript.

1) There is one critical gap in the knowledge that is provided by the data in this paper. GLaz appears to be an important lipid transport protein that moves some forms of lipid from astrocytes to neurons. This protein appears to be important for dendrite volume. Likewise, the neuronal expressed protein LpR1 also appears to be important for dendrite growth and is localized to endosomal compartments in neurons. GLaz directly binds to LpR1. Exactly how GLaz gets from astrocytes to endocytic compartments in neurons is not identified. Presumably there is a receptor for this protein-lipid complex on neurons that then guides lipid laden GLaz into endocytic compartments? It would be useful to identify this neuronal surface protein (if it exists) and show that GLaz is in fact endocytosed by neurons.

2) The authors claim that LpR1-short in neurons is required for specific types of lipid recruitment (page 11, lines 219-220) but do not provide data that supports this conclusion. The MALDI-imaging data is focused on phospholipids, DAG and TAG and all of these are reduced with knock down of LpR1 and it is not clear if this reduction is localized to a specific cell type (insufficient resolution to determine this). So while it is clear that the expression of LpR1-long results in aberrant dendrite development, it cannot be concluded from the information provided that this is due to a recruitment of specific lipids by LpR1-short. It would be relatively easy to determine what types of lipids preferentially bind LpR1.

3) A major component of lipid droplets is esterified forms of cholesterol. Presumably these would be delivered to neurons as cholesterol by astrocyte GLaz. It would be beneficial to show cholesterol and cholesterol esters in the MALDI-imaging results as CE is a major component of lipid droplets while phospholipids are relatively minor components.

Reviewer #1 (Remarks to the Author):

Brain specific lipoprotein receptors

Yin et al analyze lipid shuttling in the Drosophila nervous system. They report that the lipocalin Glial Lazarillo is secreted by astrocytes and interacts with an Apolipoprotein D homolog expressed by neurons. The topic is certainly interesting but the present study is not based on solid data and in several cases, even ignores the state of the art. Instead, the authors mix data obtained for mammalian models and Drosophila in a not permitted manner.

1. It starts in the introduction where the claim is made that astrocytes secrete apolipoproteins - none of the Drosophila references given shows any of such data.

We thank the reviewer's comments. To clarify our statement, we added "mammalian" in Line 46. Following the reviewer's suggestions, we also modified citations throughout the Introduction section.

Strangely the work of Brankatschk (eLIFE, 2014) is not even mentioned in the paper. Brankatschk et al. report that LpR1/2 is expressed primarily in glial cells in the larval brain (Brankatschk M, Dunst S, Nemetschke L & Eaton S (2014). Delivery of circulating lipoproteins to specific neurons in the Drosophila brain regulates systemic insulin signaling. Elife 3, 1–19.).

We thank the reviewer's suggestion. However, we want to clarify that the study by Brankatschk et al, eLife, 2014, did not show LpR1/2's expression in glia. The paper described the expression of LRP1/2, which is a different set of lipid binding proteins that has functions in blood-brain-barrier glia.

In fact, there are no studies on LpRs' distribution in the Drosophila CNS published by other groups. In this study, we provide multiple pieces of evidence to support the neuronal expression of LpR1, including the RNA-seq analyses on both adult and larval fly CNS, FISH experiments and expression pattern of GAL4 under the control of LpR1's endogenous promoter (Fig. 1, 2 and Supplementary Fig. 2).

2. Likewise, the work of McFerrin et al., 2017 is not mentioned - which would be important in the context of lipid droplet formation (page 2).

We were not able to find the specific reference suggested by the reviewer, McFerrin et al., 2017, in PubMed. To introduce the general background on lipid droplets, we included the review by Walther and Farese, 2012 (Line 77, 295).

The authors focus on LpR1 and the Apolipoprotein D (APOD) homolog encoded by the Drosophila gene GLaz. They claim (page 3) that in mammals APOD is the most abundant type of apolipoprotein secreted by glia. Looking up the reference and the excellent data base in the Barres lab indeed shows minor expression of APOD in human astrocytes (comparable to neurons and less than endothelial cells) but the most dramatic

expression is in oligodendrocytes (almost 10x more). Why are the claim and the data so far apart?

We believe the reviewer is referring to our statement: “APOD as the most abundant type of apolipoprotein secreted by astrocytes”. This statement is consistent with the results we obtained from analyzing RNA-seq data published by Ben Barres’ lab (Supplementary Fig. 6). We specifically compared the expression level of APOD with that of other types of apolipoproteins in astrocytes, not in all brain cells.

Upon saying that, we do agree with the reviewer that the same data set also showed the expression of APOD in neurons and oligodendrocytes. In this revision, we amended our statement to the following (Line 105-108):

“Notably, recent transcriptome profiling of human brain cells demonstrated APOD’s expression in neurons, endothelial cells, oligodendrocytes and astrocytes. In particular, APOD is the most abundant type of apolipoprotein expressed in human astrocytes (Zhang et al., 2014).

3. The manuscript starts with a lengthy description of the genomic organization and expression analysis of the two isoforms of the Drosophila LpR1 gene (5 pages). This is not what I would have expected in a Nature communications paper. Moreover, several of the experiments are way below standard in the Drosophila field. Expression analysis is either in silico or in dissociated cells! The fly brain is so small that single cell resolution is easily possible even in FISH experiments.

The reviewer raised general concerns about the experimental approaches we used in our study, specifically the in-silico analysis of RNA-seq datasets and quantitative FISH on dissociated cells. We believe these approaches greatly complement classic Drosophila genetics, with RNA-seq analyses generating transcriptome profiles in many cell types, while the exon-mapping in combination with the qFISH studies reveal and validate the isoform specific expression of LpR1 in different types of neurons.

Although FISH experiments have been performed in many model systems, quantitative FISH in whole mount Drosophila brains has been traditionally challenging. To quantify isoform-specific transcripts in a defined cell type, we established this qFISH protocol using RNAScope technology and dissociated neurons. This method allows isoform-specific probes to bind to the transcripts efficiently and offers sufficient optical resolution for the quantifications shown in Figures 1 and 2.

4. In fact, the CRISPR technology is well established in the field and insertions of small tags into the endogenous genes are easily feasible and must be done to demonstrate expression of the different proteins. Localization studies using overexpression constructs are highly questionable. Likewise, the notion that the two different LpR1 isoforms (short and long) show different localization and thus possibly different function calls for isoform specific mutants which are quite easy to generate in Drosophila.

We agree with the reviewer that the endogenously tagged proteins are superior compared to the overexpression constructs in demonstrating protein localization. However, due to the complex protein structure and modification patterns in the N-terminus of the LpR1 receptor, we were unable to identify suitable locations to generate isoform-specific knock-in lines using the CRISPR/Cas9 technique. These lines could take more than 6 months to establish and validate even without the complications mentioned above. In addition, it is also challenging to obtain cell-specific labeling of the long vs. short isoforms through endogenous tagging, which would limit the potential benefit of the knock-in lines.

To address the reviewer's concern, we generated additional transgenic lines expressing either the LpR1-long or short isoform with a small HA tag and is directly driven by a LNV specific-enhancer. These lines are also generated by site-specific integration to ensure the transgenes are inserted in the same chromosomal location. This approach allows us to demonstrate isoform specific localization of LpR1 at a comparable expression level. The results are similar to the ones we obtained by the Gal4-UAS driven expression of the GFP-tagged LpR1 isoforms. We included the representative images in new Supplementary Figure 3b. In addition, we modified our statement in the Results section to acknowledge the limitation of our current approach (Line 208-213):

“It is possible that our results generated through these overexpression studies may not faithfully represent the endogenous LpR1 distribution. However, the endosomal localization of the LpR1-short-GFP is consistent with previous findings in *Drosophila* and other insects, suggesting that the brain-specific short-isoforms of LpRs are endocytic receptors, similar to the mammalian LDLRs (Parra-Peralbo and Culi, 2011; Van Hoof, Rodenburg and Van der Horst, 2002; Dantuma et al., 1997; Beffert, Stolt and Herz, 2004).”

We also agree with the reviewer's comments regarding the importance of the isoform-specific mutants. In this revision, we generated two sets of gRNA constructs targeting the isoform-specific exons and performed CRISPR/Cas9 mediated mutagenesis specifically in LNVs. The results support our conclusions that LpR1-short is the main isoform that expresses in neurons and is required for LNV dendrite development. These results are included in new Figure 4 and described in the Results (P11-12).

5. Line 158 The MiMIC insertion used by the authors is not an enhancer trap line! The analysis of the expression pattern is not convincing.

We thank the reviewer for this comment and corrected our statement in the text (Line 149-152):

“To examine the general expression pattern of LpR1, including all long and short isoforms, in the larval brain, we obtained the LpR1-MI14131-TG4.1 line (LpR1-TG4), which contains a Gal4 element inserted in the intronic region between exon 12 and 13 and is under the control of the endogenous LpR1 promoter (Nagarkar-Jaiswal et al., 2015; Lee et al., 2018) (Fig. 2a).

In addition, we would like to point out that the LpR1-Trojan Gal4 line generated by the MiMIC insertion in the coding intron allows the expression of GAL4 under control of the endogenous promoter of LpR1, which we validated using FISH experiments. Over 95% of GFP⁺ cells labeled by the LpR1-Trojan Gal4 also show a detectable level of LpR1 expression using either the LpR1-long or LpR1-short probe (Fig. 2c).

6. Line 163 Figure 2d - what is shown here. Costaining with elav or Repo should have been done.

We performed the Repo staining following the reviewer's suggestion. The LpR1-Gal4 line (LpR1-TG4) labels a large number of cells in the larval CNS and shows a lack of coexpression with the glia marker Repo. These results are presented in the revised Supplementary Figure 2a.

7. Line 192 claims that the results are consistent with previous findings in the Drosophila PNS but no references are given to support the claim.

We would like to clarify that the references we cited address LpR1's expression and function in general peripheral tissues, not specifically the peripheral nervous system (PNS). For this claim, we provided supporting references. We have also revised the relevant sections in the main text to clarify how our results are consistent with, and complemented by, these cited studies (Lines 262-268):

“ Combined with the expression analyses using RNA-seq and qFISH, our isoform-specific genetic manipulations clearly demonstrate the neuronal-specific expression of LpR1-short and its function in supporting dendrite development and synaptic activities in neurons. These results are complementary to previous findings in Drosophila peripheral tissues, where the long-isoform of LpR1 recruits lipids through an endocytosis-independent mechanism and requires lipid transfer particle (LTP)-facilitated cell surface lipolysis (Parra-Peralbo and Culi, 2011; Van Hoof, Rodenburg and Van der Horst, 2002; Dantuma et al., 1997).”

8. Lines 229-244 Do the authors claim that lipid droplets are in neurons? How should this be explained mechanistically? The authors use elav-Gal4 to target neurons, nsyb-Gal4 would be the cleaner driver. The finding is that elav-Gal4 driven RNAi based suppression of LpR1 results in fewer lipid droplets.

We thank the reviewer for the comments and clarified our statements in the main text. In the CNS, lipid droplets are only found in glia. In the larval optic neuropil, the lipid droplets we observed are within the glia surrounding the neuronal processes. Here are the relevant sections included in the revised main text (Line 274-278, Line 285-287 and Line 291-294):

“ Previous immunohistochemistry and electron-microscopy (EM) studies of the Drosophila brain have shown that lipid droplets are only found in glia and are strongly influenced by the neuron-glia lipid trafficking and metabolic coupling (Kis et al., 2015; Bailey et al., 2015; Liu et al., 2015; Liu et al., 2017). Thus, we use the lipid droplet density as a parameter to assess the lipid content in the brain.

...To examine the effect of LpR1 deficiency on the brain lipid content, we performed Nile red staining and quantified the lipid droplet density specifically in the glia of the LON region through 3D reconstruction (Fig. 5c).

... Here, we also observed a significant reduction of brain lipid droplet density (Supplementary Fig. 4), suggesting that neuronal expression of LpR1 has a non-autonomous effect on glial lipid storage and contributes to the general maintenance of lipid content and homeostasis in the brain.”

Regarding the reduced lipid droplet density in the neuronal knock-down of LpR1 (Supplementary Fig. 4), our interpretation is, when the neuronal LpR1 is reduced by RNAi knock-down, the neuron's ability to uptake lipid is compromised, which affects the general lipid homeostasis in the brain and leads to reduced lipid droplet density in glia. Similar observations were made in the adult *Drosophila* eye, where neuronal manipulation of lipid-related molecules led to changes in glial lipid droplet density non-autonomously (Liu et al, Cell 2015). Since LpR1 is largely expressed in neurons (Fig. 2, Supplementary Fig. 2), this result is consistent with the LpR1 mutant phenotype we observed (Fig. 5).

In regard to the reviewer's comment about nSyb-Gal4 being a cleaner driver than elav-Gal4, we believe elav-Gal4 is appropriate to use in this set of experiments. Not only has elav-gal4 been widely used as a pan-neuronal enhancer Gal4 line over the past couple of decades, studies indicate that it has a high expression in the developing fly nervous system, in contrast to the nSyb-Gal4 line, which labels differentiated, mature neurons more efficiently (Yao and White, Journal of Neural Chemistry, 1994).

9. Line 248 MALDI-TOF MS experiments allow to visualize lipids in adult brain sections. Yes, but this is not done, the Figure shows head sections! and the head is full of very lipid rich fat body which might smear across the section and ruin the MALDI-TOF MS data or might obscure the data due to poor spatial resolution. The observed 10 fold decrease in TAG-content could be due to the fat body and the consequences of disruption of the global lipid metabolism, which would be expected in null mutants, I guess.

We thank the reviewer for the careful evaluation of our results. We modified the text to indicate that the MALDI imaging experiments are performed in the head sections (Line 300).

To address the concern raised by the reviewer regarding fat body contamination, we tested dissected fly brain tissue for MALDI imaging. However, without the protective head case, fly brains easily lose their shape and integrity during OCT embedding and frozen sectioning, and are not suitable for the protocol we are currently using. In addition, it is worth noting that, although we cannot completely exclude the possibility of some of the sections being contaminated by non-brain tissues, it is unlikely that the differences we observed were consistently produced by fat bodies. We would like to direct the reviewer's attention to the quantifications we presented in Supplementary Figure 5, which showed results from multiple biological repeats and statistically significant differences between the two genotypes. We also described in the Methods section that only signals from the brain region are used for quantification.

To justify our methods and acknowledge the potential limitations, we included the following statement (Line 307-309):

“Although MALDI-imaging allowed us to directly visualize a broad spectrum of lipid species, using this approach to analyze the small fly head sections has certain limitations, such as the low spatial resolution, variable detection sensitivity and potential contamination from non-brain tissues.”

From the images, I would say just the phospholipids - and not all of them - are really different in the brain. Those are membrane lipids, suggesting more membrane or depending on the type of phospholipid a different membrane composition (but I am not an expert)- not a storage phenotype.

In regard to the changes in different lipid species, please see our quantifications presented in Supplementary Figure 5. Additionally, we agree with the reviewer that LpR1 mutant does not exhibit a simple lipid storage phenotype. We have modified our language in the Results section (P13-15), removed the statement related to lipid storage and described the LpR1's function as "contributes to the general maintenance of lipid content and homeostasis in the brain".

10. Line 264/265 Previous studies indicate that, in both vertebrate and invertebrate CNS, neuronal lipid uptake and recycling rely heavily on neuron-glia interactions. None of the reference listed shows this for invertebrates, likewise, one sentence later, Ma et al., never showed that astrocyte derived apolipoprotein has a critical function in synapse formation and function.

We thank the reviewer for pointing out the issues with our citations. In the revised manuscript, we modified the citations to include Drosophila references, Volkenhoff et al., 2015, Liu et al., 2015 and Liu et al., 2017. We have also removed the inappropriate reference, Ma et al. 2016, as suggested by the reviewer.

11. The analysis of GLaz continues in the same manner. Localization is studied using an uncharacterized (and not even shown) construct where a GLaz enhancer drives a GFP-GLaz fusion. The endogenous gene must be tagged or antibodies should be generated. Again, I did not see any link to mutants.

As suggested by the reviewer, we added the missing information in the main text to describe the loss-of-function mutants of GLaz and the GLaz enhancer-driven GFP-tagged GLaz protein, both of which are published reagents that have been previously described. The relevant texts are on Page 17, Line 368-369 and :

"We performed genetic analyses using a loss-of-function mutant of GLaz, GLaz^{Δ2}, which contains a null mutation and does not produce the GLaz transcript (Sanchez et al., 2006). ...To determine the distribution of GLaz protein in the larval brain, we examined a transgenic line expressing a GFP-tagged GLaz protein driven by a GLaz enhancer (GLaz>GLaz-GFP) (gifted by Dr. Maria D. Ganfornina). Previous studies demonstrated that the expression of GLaz>GLaz-GFP rescues the loss-of-function mutant, and potentially serves as an indicator for the endogenous level and distribution of the GLaz protein (Sanchez et al., 2006; del Cano-Espinel et al., 2015)."

Lines 334-338: How can the authors claim that "GLaz-HA is expressed in astrocytes and modified by glycosylation". They used al¹m-Gal4 to express the protein in astrocytes.

We thank the reviewer for pointing out the issues with our statement and made clarifications in our revised manuscript. The sentence (Line 408-410) now reads,

“ Consistent with previous publications, we detected two bands around 27-30kDa with anti-HA antibody, suggesting that GLaz-HA is correctly expressed and modified by glycosylation, , similar to the modification reported in human APOD (Supplementary Fig. 8a) (Ruiz et al., 2012).”

In summary, I don't think that this paper reports anything concrete about lipid uptake into neurons. The authors do not show that the LDs are in neurons, neither can they deduce this from the MALDI imaging experiments. They ignore published data (Brankatschk,2014) and link references to claims that are not supported in these studies.

We thank the reviewer's evaluation of our work and constructive comments, which we believe we have fully addressed in our revision by new experiments and adjustment of our statements. We also want to point out, as stated in our responses to points #1 and #2, we did not miss the critical reference as the reviewer suggested. We hope those issues have been clarified here.

Reviewer #2 (Remarks to the Author):

The manuscript by Yin et al. reports that trafficking of lipids from glial cells to neurons is important for activity dependent dendrite growth in the Drosophila central nervous system. A novel contribution of this study is to identify mechanisms of trafficking of lipids from glia to neurons and to show that activity regulates lipid shuttling through upregulation of the lipophorin receptor LpR1. The authors make several important discoveries regarding lipid shuttling in the central nervous system, including the identity of a putative glial derived lipoprotein Glaz that can promote dendrite morphogenesis, and LpR1 as a receptor for the lipoprotein on LNV dendrites. The developmental role for Glaz is particularly significant because prior studies had hypothesized developmental roles, but this had not yet been demonstrated. The findings should provide a map for further studies in mammalian systems. Evidence for functional interactions between Glaz and LpR1 are compelling. However, some of the conclusions lack strong support, dampening my enthusiasm for the manuscript somewhat without clarification of these concerns.

Major comments:

1. Line 182-185. One major conclusion of the manuscript, that different isoforms of LpR1 are localized differently in neurons, is limited because it appears to derive from UAS-Gal4 overexpression. The details of this are hard to find in the results, but in the methods I could only find UAS lines described. It was also not clear whether the transgenes were inserted into the same location in the fly genome. If not, this would make it hard to compare localization directly because the transgenes could be expressed at vastly different levels. Because overexpression could affect protein localization, the conclusion that LpR1-short and LpR1-long display distinct cellular localization in LNVs has several caveats. Another limitation to localization and rescue experiments is that only a single UAS line is used for overexpression of each of the isoforms.

We agree with the reviewer that there are significant limitations of using overexpression constructs to study protein localization. However, due to the complex protein structure and modification pattern at the LpR1 N-terminus, we are unable to generate endogenously tagged

short vs. long isoforms of LpR1 during the revision. To address the reviewer's concern, we generated additional transgenic lines expressing either the LpR1-long or short isoform with a small HA tag, and are directly driven by a LNV-specific enhancer. These lines are also generated by site-specific integration to ensure the transgenes are inserted in the same chromosomal location. This approach allows us to demonstrate isoform-specific localization of LpR1 at a comparable expression level. The results are similar to the ones we obtained by the Gal4-UAS driven expression of the GFP tagged LpR1 isoforms. We included the descriptions of these new lines in the Results section (P10) and the representative images in new Supplementary Figure 3b. In addition, we modified our statement to acknowledge the limitation of our current approach (Line 208-213):

"It is possible that our results generated through these overexpression studies may not faithfully represent the endogenous LpR1 distribution. However, the endosomal localization of the LpR1-short-GFP is consistent with previous findings in *Drosophila* and other insects, suggesting that the brain specific short-isoforms of LpRs are endocytic receptors, similar to the mammalian LDLRs (Parra-Peralbo and Culi, 2011; Van Hoof, Rodenburg and Van der Horst, 2002; Dantuma et al., 1997; Beffert, Stolt and Herz, 2004)."

2. line 212-213 it is proposed that LpR1-short isoforms may rescue by recruiting specific types of lipids for establishing synaptic connections. There is no strong evidence that this is the case, especially given that long and short isoforms can fully rescue the phenotype. The difference being that long isoforms cause an ectopic dendrite growth. Furthermore, because rescue is obtained with both short and long isoforms, isoform identity seems not to be important for this aspect of morphogenesis. The rescue by both also challenges the notion that differential localization shown in figure 3 is important for morphogenesis. In the same section the authors suggest (on lines 218-220) that isoform specific expression of LpR1-short ensures proper synaptic connectivity, but the only difference here is that LpR1-long leads to exuberant dendrites growing off in one direction. There is no evidence that this difference in morphology leads to different synaptic connectivity either on those exuberant dendrites or the normal ones.

We thank the reviewer for raising this important point. To address the reviewer's concerns, in this revision, we performed calcium imaging for the genetic rescue experiments to further evaluate the functional differences of the long and short isoforms of LpR1. Although the gross morphology of the LNV dendrite can be rescued by both isoforms, the long isoform's expression failed to rescue physiological deficits, while the short isoform's expression showed a full functional rescue. We presented this new set of data in revised Figure 3d and describe the results in the Main text (P11), with the statement below (Line 234-240):

"This set of comparisons made between the LpR1-long vs short isoforms reveals the strong influence of isoform specificity on the receptor's function. Importantly, replacing the native LpR1-short with the exogenously expressed long isoform in LNVs leads to exuberant dendrite growth and severely dampened physiological responses. Our results suggest that, although both short and long isoforms of LpR1 are capable of recruiting lipids to support dendrite growth, expression of LpR1-short in neurons is essential for the proper establishment of synaptic connectivity during development."

3. Line 243, it was not obvious how neuronal expression of LpR1 contributes to lipid droplet storage, presumably in glia? Would be useful to have some clarification of this.

We thank the reviewer for raising this important point. Regarding the reduced lipid droplet density in the neuronal knock-down of LpR1, our interpretation is, when neuronal LpR1 is reduced, the neuron's ability to uptake lipid is compromised, which affects the general lipid homeostasis in the brain and leads to reduced lipid droplet density in glia. Similar observations were made in the adult *Drosophila* eye, where the neuronal manipulation of lipid-related molecules led to the changes in glial lipid droplet density non-autonomously (Liu et al, Cell 2015). Since LpR1 is largely expressed in neurons (Fig. 2, Supplementary Fig. 2), this result is consistent with the LpR1 mutant phenotype we observed (Fig. 5).

Following the reviewer's suggestion, we modified the statement in the main text (Line 274-278, Line 285-287 and Line 291-294):

“ Previous immunohistochemistry and electron-microscopy (EM) studies of the *Drosophila* brain have shown that the lipid droplets are only found in glia and are strongly influenced by the neuron-glia lipid trafficking and metabolic coupling (Kis et al., 2015; Bailey et al., 2015; Liu et al., 2015; Liu et al., 2017). Thus, we use lipid droplet density as a parameter to assess lipid content in the brain.

...To examine the effect of LpR1 deficiency on brain lipid content, we performed Nile red staining and quantified the lipid droplet density specifically in the glia of the LON region through 3D reconstruction (Fig. 5c).

... Here, we also observed a significant reduction of brain lipid droplet density (Supplementary Fig. 4), suggesting that neuronal expression of LpR1 has a nonautonomous effect on glial lipid storage and contributes to the general maintenance of lipid content and homeostasis in the brain.”

Minor points:

1. In the abstract the authors claim that lipid transport mediated by Glaz is bidirectional. What is the evidence for this?

We thank the reviewer for raising this valid point. To avoid confusion, we removed “bidirectional” from the abstract and included the following statement in the discussion (Line 489-494):

“ Recent studies in adult *Drosophila* compound eye have demonstrated that GLaz is involved in lipid transfer from neurons to glia (Liu et al, 2017), while we identified the interaction between GLaz and LpR1 and the possible role for GLaz in delivering lipids to neurons. Therefore, GLaz appears to be involved in both sides of the lipid trafficking and potentially serves as the key molecular target for studies related to the bidirectional neuron-glia lipid shuttling in neuronal development and plasticity.”

2. In lines 103-105 the authors imply that they have identified the APOD receptor as LpR1. The authors have identified the receptor for the APOD homolog Glaz. Although the evidence for conservation cited in the discussion is promising, it is still not clear whether this is going to generalize.

We agree with the reviewer that our previous statement is not accurate. We amended the statement in Line 108-111 as:

“Given the close connections between APOD, stress resistance, neurological disorders and aging in both human and *Drosophila* (Muffat, Walker and Benzer, 2008; Ruiz et al., 2013), the identification of LpR1-short as the neuronal receptor of GLaz, the close homologue of human APOD, has broad implications.”

3. Is the requirement for lipid shuttling dendrite specific or is there any requirement for axon morphogenesis? These data may not be available and it is not an essential new experiment, but if the authors have any relevant information on this it could be interesting to comment or speculate on.

We thank the reviewer for raising this important point. Based on our observations, we think LpR1's function is largely required for dendrite morphogenesis, but much less so for axon development. We have previously shown that, while there is a significant reduction in LNV dendrite volume, the axon morphology is not affected in LpR1 knock-down flies (Supplementary Figure 6 from Yin et. al. 2018). This distinction between the axonal and dendritic compartments suggests that, compared to the axon, LNV dendrites are more sensitive to reduced levels of LpR1, possibly due to a higher demand for localized lipid uptake during development. This notion is consistent with LpR1's dendritic localization (Fig. 3a) and the recruitment of GLaz into the vicinity of dendritic arbors (Fig. 7c). However, since these observations are still disconnected and need additional supportive evidence, we will not speculate on the implications in this manuscript.

4. In reporting the results on the expression of LpR1 the authors go through a very detailed and rigorous description of isoform specific expression then move on to a less specific Gal4 line. The line and the data are not necessarily problematic, but this contributes very little coming after the RNA-seq and qFISH data. Perhaps starting with the Gal4 data then moving to the finer analysis would be more compelling.

We thank the reviewer for the suggestion. The Results section and the corresponding Figure 2 have been reorganized as suggested by the reviewer.

TRACT technique is not in wide use so would be informative to have more description of the technique either in methods or results. Also, it is not clear that controls were performed (presumably standard control would be lack of lexA driver).

We thank the reviewer's suggestion. We included an additional description in the Methods and the control image in Supplementary Figure 6.

5. Figure 6c. Was the RNAi screen dataset corrected for multiple comparisons?

For Figure 6c, the statistics are performed using student's t-test by comparing the individual RNAi knock-down to the control group. Therefore, they are not corrected for multiple comparisons. We included this information in the Methods section.

6. Figure 7: Are these images confocal projections? If so, what is the location of GLaz-GFP relative to dendrites in 3D?

We thank the reviewer for raising this important point. In this revision, we included additional imaging data to illustrate the spatial relationship between GLaz and the LNV dendrite in 3D. Upon closer inspection, we found numerous GLaz-GFP punctate localized near or on the surface of the LNV dendrites, suggesting that secreted GLaz protein could be captured by LpR1 through direct interactions and perform the lipid transfer at the cell surface without being internalized. We described the results in the main text (Line 395-405) and presented the new data as Figure 7c and Supplementary Figure 7.

Rab1DN results indicate a defect in trafficking in astrocytes, however the authors conclude based on these results alone that GLaz is secreted from astrocytes. What is their justification for this conclusion?

We thank the reviewer for pointing out the missing information and the inaccuracy in our statement. We revised the main text on P17-18 and provided information on the GLaz>GLaz-GFP transgenic line, which expresses a GFP-tagged GLaz protein driven by a GLaz enhancer (GLaz>GLaz-GFP). In addition, we clarified the purpose of the astrocyte secretion blocking experiments and provided our justifications on using the expression of the GLaz>GLaz-GFP as an indicator for the GLaz protein (Line 391-394):

“By comparing the GLaz>GLaz-GFP distribution pattern between a control group and one in which secretion from astrocytes is blocked, (Fig. 7C), we concluded that GLaz-GFP is secreted by astrocytes, consistent with the expected localization of the endogenous GLaz protein.”

7. Line 378-379 of discussion – authors have not addressed LpR2, so why do they generalize the results to both LpRs?

We thank the reviewer's comments. Our previous study indicates that LpR1 and LpR2 are similar in their contributions to dendrite morphogenesis. But since this study focuses on LpR1, we agree with the reviewer and amended our statements to discuss only LpR1.

Reviewer #3 (Remarks to the Author):

This is a very interesting manuscript that identifies two major players that regulate astrocyte to neuron lipid transport that appear to play important roles in dendrite development. The experiments are elegantly designed and utilize a number of over expression, known down, and protein labeling techniques to describe cellular localization, and confirm the involvement of LpR1 and GLaz in astrocyte to neuron lipid transport. The images presented in this paper are excellent visual representations of the experimental results. Likewise the manuscript itself is well written and makes a convincing case for the involvement of these two proteins in lipid transport and dendrite development.

There are several relatively minor deficiencies that I believe would improve the manuscript.

1) There is one critical gap in the knowledge that is provided by the data in this paper. GLaz appears to be an important lipid transport protein that moves some forms of lipid from astrocytes to neurons. This protein appears to be important for dendrite volume. Likewise, the neuronal expressed protein LpR1 also appears to be important for dendrite growth and is localized to endosomal compartments in neurons. GLaz directly binds to LpR1. Exactly how GLaz gets from astrocytes to endocytic compartments in neurons is not identified. Presumably there is a receptor for this protein-lipid complex on neurons that then guides lipid laden GLaz into endocytic compartments? It would be useful to identify this neuronal surface protein (if it exists) and show that GLaz is in fact endocytosed by neurons.

We thank the reviewer for raising this important question. Our initial hypothesis is that LpR1-short, an endocytic receptor, could facilitate or mediate the internalization of the GLaz protein into the neuron. However, our new data support a different model. In this revision, we included additional imaging data to illustrate the spatial relationship between GLaz and the LNV dendrite using 3D reconstruction. Upon closer inspection, we found numerous GLaz-GFP punctate localized near or on the surface of the LNV dendrites, ~51 per brain lobe, while very few GFP punctate were observed within the soma, only one found in 9 brains, suggesting that secreted GLaz protein may be captured by LpR1 through direct interactions and perform lipid transfer at the cell surface without being internalized. We described the results in the main text (Line 395-407) and presented the new imaging data and quantification as Figure 7c and Supplementary Figure 7.

2) The authors claim that LpR1-short in neurons is required for specific types of lipid recruitment (page 11, lines 219-220) but do not provide data that supports this conclusion. The MALDI-imaging data is focused on phospholipids, DAG and TAG and all of these are reduced with knock down of LrP1 and it is not clear if this reduction is localized to a specific cell type (insufficient resolution to determine this). So while it is clear that the expression of LpR1-long results in aberrant dendrite development, it cannot be concluded from the information provided that this is due to a recruitment of specific lipids by LpR1-short. It would be relatively easy to determine what types of lipids preferentially bind LpR1!.

We agree with the reviewer that we did not provide direct experimental evidence to support the claim that LpR1-short is required for recruiting specific types of lipid(s). We reached this conclusion mainly based on our expression profiling and genetic rescue experiments. To strengthen our claims, in this revision, we performed calcium imaging for the genetic rescue experiments to further evaluate the functional differences of the long and short isoforms of LpR1. Although the gross morphology of the LNV dendrite can be rescued by both isoforms, the long isoform expression failed to rescue the physiological deficit, while the short isoform expression showed a full functional rescue. This new result is presented in Figure 3d and described on P11. To clarify our claims, we included the following statement (Line 262-268):

“ Combined with the expression analyses using RNA-seq and qFISH, our isoform-specific genetic manipulations clearly demonstrate the neuronal-specific expression of LpR1-short and its function in supporting dendrite development and synaptic activities in neurons. These results are complementary to previous findings in *Drosophila* peripheral tissues, where the long-isoform of LpR1 recruits lipids through an endocytosis-independent mechanism and requires lipid transfer

particle (LTP)-facilitated cell surface lipolysis (Parra-Peralbo and Culi, 2011; Van Hoof, Rodenburg and Van der Horst, 2002; Dantuma et al., 1997).”

With the new information generated during the revision, including the calcium imaging (Fig. 3d), isoform-specific mutagenesis (Fig. 4) and detailed evaluations of GLaz-GFP distribution (Fig. 7c and Supplementary Fig. 7), we now have a modified model to explain how different isoforms of the LpR1 receptor perform lipid recruitment in neurons vs. peripheral tissues. We presented this model in Figure 7e and discussed the importance of identifying the lipid cargo of GLaz in the main text (P23).

We also agree with the reviewer that our MALDI imaging approach lack spatial resolution and did not provide specific information regarding the lipid species that bind to LpR1. We acknowledge this limitation in our revised manuscript (Line 307-312):

“ Although MALDI-imaging allowed us to directly visualize a broad spectrum of lipid species, using this approach to analyze the small fly head sections has certain limitations, such as the low spatial resolution, variable detection sensitivity and potential contamination from non-brain tissues. Notably, while both of our analyses revealed a general reduction of lipid content in the LpR1 mutant, the specific lipid cargo(s) being transferred by LpR1 within the fly CNS remains unidentified.”

In addition, we agree that identifying the specific lipid species that binds to LpR1 is particularly exciting and important. But we also think it remains challenging with current technology. Due to the complex neuron-glia interactions, local concentrations and modifications of specific types of lipids are difficult measure in brain and to reproduce in a non-native environment. And the general lipidomics experiments are not be able to provide cell-type specific information, which could be critical for isoform or receptor-specific lipid recruitment. Upon saying that, a recent study (Fitzner et al, Cell Reports, Sept. 2020) analyzed cell-type and brain-region-specific lipid profiles in mice using quantitative shotgun lipidomics. Similar technical advance in lipidomics or improved MALDI imaging techniques with cellular resolution may offer us useful information in the near future.

3) A major component of lipid droplets is esterified forms of cholesterol. Presumably these would be delivered to neurons as cholesterol by astrocyte GLaZ. It would be beneficial to show cholesterol and cholesterol esters in the MALDI-imaging results as CE is a major component of lipid droplets while phospholipids are relatively minor components.

We thank the reviewer for the great suggestion. Cholesterol and cholesterol ester were not detected in our previous MALDI imaging dataset. During this revision, our collaborators at CUNY performed additional MALDI imaging using different chemical treatments (described in the Methods) and successfully obtained readings from the specific spectrums corresponding to cholesterol, although cholesterol ester remained undetected. These new results, presented in Supplementary Figure. 5, indicate that the LpR1 mutants also show a reduction in the CNS cholesterol level as compared to the control flies.

REVIEWER COMMENTS

Reviewer #1 (Remarks to the Author):

The authors addressed several of the comments and I appreciate the inclusion of isoform specific knockout experiments. However, I have to confess that I still have concerns that are detailed below. Let me start by apologizing for inserting a wrong reference, it should have been Cabirol-Pol et al 2017. Although the current text improved there is still a general overstatement and over-interpretation of the data.

Starting with the introduction the authors continue to frequently mix mammalian and *Drosophila* biology. For example, a sentence such as:

"Intriguingly, apolipoproteins are among the most abundant secretory factors that are being produced and released by mammalian astrocytes (Allen and Eroglu, 2017; Mahley, 2016), a group of glial cells with complex morphology and highly branched structures that are intimately associated with synapses (Muthukumar, Stork and Freeman, 2014; Allen and Lyons, 2018), suggesting a critical role for glia-derived lipoprotein and their lipid cargos in synapse formation and function (Stork et al., 2014; Boyles et al., 1985; Wang and Eckel, 2014)."

This sentence contains several misleading cross-references. Indeed, mammalian astrocytes are complex cells that can be associated with synapses, *Drosophila* astrocyte-like cells are not (MacNamee et al 2016). In the Stork et al., paper the word lipoprotein does not occur and no evidence for the made assumption "glia-derived lipoprotein and their lipid cargos in synapse formation and function" is presented or even discussed. This type of imprecision is detectable throughout the manuscript.

In their response, the authors argued that CRISPR-based gene tagging is too time consuming. In my view this an invalid argument. An analysis on the cellular/subcellular localization cannot be based on overexpression constructs and should be foundation of a solid study.

Please explain the images showing that LpR1 is not expressed in glia. It is unclear what sup. Figure 2 shows, it looks like a frontal view on the brain surface. It would be better to focus on the relevant glial cells, the astrocytes, and show a corresponding section of the brain.

Antibodies to label astrocytes are available.

Figure 5 d What exactly is shown here? To examine the effect of LpR1 deficiency on the brain lipid content, we performed Nile red staining and specifically quantified the lipid droplet density in the glia of the LON region through 3D reconstruction (Fig. 5c). How is it possible to define where glia is? From the image it appears that the LNV neuron contains lipid droplet but lipid droplets should be in glial cells? Please explain. The same applies for Figure 6f.

The stock mentioned in Materials and Method (GLaz (RNAi), BDSC: 67728) is not a GLaz RNAi stock.

The analysis of Glaz is not state of the art. Instead of using a tool that was developed in the lack of alternatives in 2006, one should use today's tools that allow detection of endogenous proteins. There is even a GFP-converted MiMIC available in Bloomington which allows to analyze the endogenous expression pattern. Studies on subcellular localization using Gal4 directed expression are not valid.

The sentence: "this result supports the model that the glia-derived GLaz is recruited by the lipoprotein receptors onto the surface of LNV dendrites, where it delivers lipid cargo without being internalized through the endocytic pathway (Fig. 7e)." is not supported by the ectopic expression data.

I do not understand the logic the following experiment is described.

"To determine whether GLaz directly binds to the LpR1 receptor, we performed co-

immunoprecipitation (Co-IP) experiments using HA-tagged GLaz expressed in astrocytes, driven by the Alrm-Gal4 driver, and GFP-tagged LpR1-short and long expressed in neurons, driven by the elav-Gal4 driver."

It should be spell out clearly that different stocks are generated, HA-tagged GLaz was loaded onto beads and subsequently these beads were used to pulldown LpR1 short (long). These blots are truly amazing and it would be nice to hear how much of the input is shown on the left panel (%) to judge the power of the approach.

Reviewer #2 (Remarks to the Author):

The authors have addressed all of my initial concerns. This is a very well-done and important study.

Point-by-point response to the reviewers' comments (NCOMMS-20-09880B)

“Brain-specific lipoprotein receptors interact with astrocyte derived apolipoprotein and mediate neuron-glia lipid shuttling” by Yin et al.

We thank the editor and reviewers for their evaluations of our work and constructive suggestions. Here we address comments from reviewer # 1 by providing amendments and clarifications in the text and citations, as well as adding new results demonstrating the endogenous expression pattern of GLaz protein (**new Supplementary Figure 8**). We hope that the editor and reviewers find this revised version now suitable for publication in Nature Communications.

Reviewer #1 (Remarks to the Author):

1. *The authors addressed several of the comments and I appreciate the inclusion of isoform specific knockout experiments. However, I have to confess that I still have concerns that are detailed below. Let me start by apologizing for inserting a wrong reference, it should have been Cabirol-Pol et al 2017.*

Response:

We thank the review for suggesting the reference, which we included in Introduction (Ref. #30, Line 87).

2. *Although the current text improved there is still a general overstatement and over-interpretation of the data. Starting with the introduction the authors continue to frequently mix mammalian and Drosophila biology. For example, a sentence such as: "Intriguingly, apolipoproteins are among the most abundant secretory factors that are being produced and released by mammalian astrocytes (Allen and Eroglu, 2017; Mahley, 2016), a group of glial cells with complex morphology and highly branched structures that are intimately associated with synapses (Muthukumar, Stork and Freeman, 2014; Allen and Lyons, 2018), suggesting a critical role for glia-derived lipoprotein and their lipid cargos in synapse formation and function (Stork et al., 2014; Boyles et al., 1985; Wang and Eckel, 2014)."*
This sentence contains several misleading cross-references. Indeed, mammalian astrocytes are complex cells that can be associated with synapses, Drosophila astrocyte-like cells are not (MacNamee et al 2016). In the Stork et al., paper the word lipoprotein does not occur and no evidence for the made assumption "glia-derived lipoprotein and their lipid cargos in synapse formation and function" is presented or even discussed. This type of imprecision is detectable throughout the manuscript.

Response:

We thank the reviewer for pointing out the issues with our citations. In this revision, we reorganized the citations throughout the main text, separated *Drosophila* and mammalian literatures, and clearly stated the model system used in each referenced work. We also formatted the citation using Nature Communications' style. Please see the tracked modifications in the main text. Regarding the specific section mentioned in the reviewer's comments, we modified the text on as below (**P3**):

*“Intriguingly, apolipoproteins are among the most abundant secretory factors that are produced and released by mammalian astrocytes^{6,7}, a group of glial cells with complex morphology and highly branched structures that are intimately associated with synapses^{6,8}, suggesting a critical role for glia-derived lipoprotein and their lipid cargos in synapse formation and function^{2,9}. This notion is supported by studies in cultured mammalian CNS neurons, where glia-derived cholesterol and phospholipids are essential for synaptogenesis^{9,10,11}. In addition, recent findings in the *Drosophila* system also indicate essential functions of glia in synapse formation and neurotransmission^{12,13,14,15}, although the link between neuron-glia lipid transport and synaptic function has yet to be established.*

- 3. In their response, the authors argued that CRISPR-based gene tagging is too time consuming. In my view this an invalid argument. An analysis on the cellular/subcellular localization cannot be based on overexpression constructs and should be foundation of a solid study.*

Response:

We appreciate the reviewer’s concern. We agree that the cellular localization of an overexpressed protein does not always reflect the endogenous pattern. We would like to clarify that we could not generate the isoform-specific CRISPR knock-in lines because of technical issues, not simply due to the time restrains. To generate isoform specific knock-in lines, we need to insert an epitope or GFP tags into the N-terminus of the LpR1 protein, where the short and long isoforms differ. Due to the complex protein structure and modification patterns in the N-terminus of LpR1, as well as the small sizes of isoform-specific exons, we were unable to identify suitable locations to insert protein tags using the CRISPR/Cas9 technique. We also have additional concerns associated with the knock-in approach. It is possible that the protein tag inserted at the N-terminus affects the folding and trafficking of the LpR1 receptor. Furthermore, the knock-in lines would label LpR1 isoforms in all cells. Without cell-specific labeling, given the broad expression of LpR1 in the larval CNS, we may not be able to visualize the localization of endogenous receptors with sufficient resolution.

Given the reasons mentioned above, to address concerns associated with the overexpression studies, in our previous revision, we generated new transgenic lines by site-specific integration. These lines express either the LpR1-short or long isoform with a small HA tag and is directly driven by a LNV specific-enhancer. Since the transgenes are inserted at the same chromosomal location and driven by the same enhancer, they are expressed at a comparable level. As shown by the results included in **Supplementary Figure 3b**, the HA-tagged short and long-isoforms showed similar cellular localization as the Gal4-UAS driven expression of GFP-tagged LpR1 isoforms (**Supplementary Figure 3a**).

Lastly, we agree with the reviewer that a protein’s cellular localization is important. Therefore, we conducted these localization studies and present the results as a part of the supplementary data (**Supplementary Figure 3**). In fact, the endosomal localization of *Drosophila* and insect LpR proteins, as well as their mammalian homologue LDLRs, have been well demonstrated in previous studies (Parra-Peralbo and Culi, 2011; Van Hoof, Rodenburg and Van der Horst, 2002; Beffert, Stolt and Herz, 2004), and our results are in agreement with those findings. Because of that, we

chose to focus on isoform-specific functions of LpR proteins in the LNvs, for which isoform specific knock-out studies (**Fig. 4**) were more informative.

To more appropriately state our claims, we amended all statements regarding the differences between the two isoforms' cellular localizations in the main text, changed the title of Supplementary Figure 3, and modified the relevant section in Results to acknowledge the limitations of our current approach (**P10**):

“It is possible that our results generated through these overexpression studies may not faithfully represent the endogenous LpR1 distribution. However, the endosomal localization of the LpR1-short-GFP is consistent with previous findings in Drosophila and other insects, suggesting that brain specific short-isoforms of LpRs are endocytic receptors, similar to the mammalian LDLRs^{33, 48, 49}”

4. Please explain the images showing that LpR1 is not expressed in glia. It is unclear what sup. Figure 2 shows, it looks like a frontal view on the brain surface. It would be better to focus on the relevant glial cells, the astrocytes, and show a corresponding section of the brain. Antibodies to label astrocytes are available.

Response:

We thank the reviewer for the comments. Following the reviewer's previous suggestions (major concern #6 in the first review), we performed anti-Repo staining to illustrate that LpR1 does not express in glia (**Supplementary Figure 2**). The LpR1-Gal4 line (LpR1-TG4) labels a large number of cells in the larval CNS and shows a lack of co-expression with the pan-glia marker Repo. Since the Repo antibody also labels astrocytes and other types of glia, we wouldn't expect to gain new results by using an astrocyte-specific antibody, merely the same lack of co-expression we observed here.

To improve the clarity of our data presentation as suggested by the reviewer, we included additional panels in **Supplementary Figure 2** in this revision:

Supplementary Figure 2a: Top: Representative projected confocal images collected from a whole mount larval brain lobe. The nuclei of glia are labeled by anti-Repo antibody. The LpR1 expressing cells are labeled by LpR1-TG4 driven expression of the membrane marker mCD8::GFP (green) and the nucleus marker redStinger (red), which does not overlap with the anti-Repo staining (gray). Bottom: Single optic sections and zoomed-in images clearly showed that none of the redStinger expressing cells are labeled by the anti-Repo antibody (grey).

5. Figure 5 d What exactly is shown here? To examine the effect of LpR1 deficiency on the brain lipid content, we performed Nile red staining and specifically quantified the lipid droplet density in the glia of the LON region through 3D reconstruction (Fig. 5c). How is it possible to define where glia is? From the image it appears that the LNv neuron contains lipid droplet but lipid droplets should be in glial cells? Please explain. The same applies for Figure 6f.

Response:

We thank the reviewer for the comments. To clarify, although we did not define glia in the LON region, since lipid droplets are only found in glia in the larval CNS, shown by both EM and confocal imaging from two previous studies (Kis et al., 2015; Bailey et al., 2015, also see the relevant images below), we assume the lipid droplets we observed in the LON are localized in glia. We introduced this background information and also amended our statement on **P13**:

“Previous immunohistochemistry and electron-microscopy (EM) studies of the Drosophila brain have shown that lipid droplets are only found in glia and are strongly influenced by neuron-glia lipid trafficking and metabolic coupling^{27, 28, 29, 55}.”

“To examine the effect of LpR1 deficiency on the brain lipid content, we performed Nile red staining and specifically quantified the lipid droplet density in the LON region through 3D reconstruction (Fig. 5c).”

kis et al., 2015

Bailey et al., 2015

Furthermore, the absence of lipid droplets in LNvs was confirmed in our study. Although the maximum projected images in Figure 5 and 6 appear to have Nile Red staining overlapping with the LNs labeled by CD8::GFP, when we examined the single optic sections in Y and Z-axis, such as the representative images shown below, we found the lipid droplets (Red) in contact with the plasma membrane but not localized inside the LNvs.

6. *The stock mentioned in Materials and Method (GLaz (RNAi), BDSC: 67728) is not a GLaz RNAi stock.*

Response:

We thank the reviewer for pointing out the typo. The correct line number is 67228. We apologize for the mistake and made corrections in this revision.

7. *The analysis of Glaz is not state of the art. Instead of using a tool that was developed in the lack of alternatives in 2006, one should use today's tools that allow detection of endogenous proteins. There is even a GFP-converted MiMIC available in Bloomington which allows to analyze the endogenous expression pattern. Studies on subcellular localization using Gal4 directed expression are not valid.*

Response:

We thank the reviewer for the suggestion. In this revision, we examined the GLaz-MiMIC-GFP line (GLaz^{MI02243-GFSTF.0}), and presented the new results in the **new Supplementary Figure 8** and described on **P18** in the main text:

“In the larval CNS, GLaz-MiMIC-GFP showed a diffused distribution pattern and was observed near the surface of LNv dendrites, similar as the results obtained through GLaz>GLaz-GFP (Figure 7b, c, Supplementary Fig. 7, 8), albeit with a less punctuated appearance. In addition, blocking secretion in astrocytes through the expression of a dominant-negative Rab1 also led to accumulation of GLaz-MiMIC-GFP in the astrocyte soma (Supplementary Fig. 8b).”

Previously, we examined the GLaz's localization and distribution using a GLaz>GLaz-GFP transgene, and presented the results in **Figure 7b, c and Supplementary Figure 7**. This transgene expresses a GFP tagged GLaz protein driven by a 1.8Kb GLaz promoter and was validated by

functional rescue experiments in prior studies (Sanchez et al., 2006; del Cano-Espinel et al., 2015). In the **revised Figure 7b**, we included a schematic diagram to clarify the components included in the transgene. Since results we obtained from GLaz-MiMIC-GFP and GLaz>GLaz-GFP largely agree with each other, we presented both sets of data in the revised manuscript to strengthen our conclusion. Please also see the response to Concern #8 below.

8. *The sentence: "this result supports the model that the glia-derived GLaz is recruited by the lipoprotein receptors onto the surface of LNV dendrites, where it delivers lipid cargo without being internalized through the endocytic pathway (Fig. 7e)." is not supported by the ectopic expression data.*

Response:

To illustrate the spatial relationship between GLaz and LNV dendrites, we performed 3D reconstruction and found numerous GLaz-GFP punctate localized near or on the surface of the LNV dendrites, but rarely within LNV soma or dendritic arbors, suggesting that secreted GLaz protein could be captured by LpR1 through direct interactions and perform the lipid transfer at the cell surface without being internalized. We described the results in the main text (**P17**) and presented these data in **Supplementary Figure 7**. The new results we obtained by examining GLaz-MiMIC-GFP localization support this hypothesis and also showed GLaz-GFP signal on the surface of the LNV dendritic arbors (**Supplementary Figure 8c**).

To address the reviewer's concern, we amended our statements on **P18** to more properly state our claims:

"Using two different GFP-tagged GLaz proteins, we confirmed that GLaz is secreted by astrocytes and is localized close to the surface of the LNV dendrite (Figure 7b, c, Supplementary Fig. 7, 8). Although additional manipulations and in vivo imaging experiments are needed to reveal how GLaz is trafficked within the synaptic region, our results suggest that astrocyte-derived GLaz is likely recruited by neuronal lipoprotein receptors onto the surface of LNV dendrites, where it delivers lipid cargo without being internalized (Fig. 7c, Supplementary Fig. 7, 8c)."

9. *I do not understand the logic the following experiment is described.
"To determine whether GLaz directly binds to the LpR1 receptor, we performed co-immunoprecipitation (Co-IP) experiments using HA-tagged GLaz expressed in astrocytes, driven by the Alrm-Gal4 driver, and GFP-tagged LpR1-short and long expressed in neurons, driven by the elav-Gal4 driver."
It should be spell out clearly that different stocks are generated, HA-tagged GLaz was loaded onto beads and subsequently these beads were used to pulldown LpR1 short (long). These blots are truly amazing and it would be nice to hear how much of the input is shown on the left panel (%) to judge the power of the approach.*

Response:

We thank the reviewer for the positive comments on our co-IP experiments and amended our statements in the main text to clarify the experimental procedures (**P19**):

"To determine whether GLaz directly binds to the LpR1 receptor, different Drosophila stocks, Alrm-Gal4>UAS-GLaz-HA and elav-Gal4>UAS-LpR1-short/long-GFP, were generated for the co-immunoprecipitation (Co-IP) experiments. HA-tagged GLaz protein was extracted from 30

larval brains and loaded onto anti-HA magnetic beads. These beads were subsequently used to pulldown protein extracts containing LpR1-short/long-GFP. 10% of input protein extracts from each genotype were included as positive controls.”